EMBO
reports

# Type-2 innate signals are dispensable for skeletal muscle regeneration and pathology linked to Duchenne muscular dystrophy

Melina Messing [iD], Marine Theret [iD], Michael R Hughes [iD], Jiaqi Wu, Omar Husain Syed, Fang Fang Li [iD], Yicong Li, Fabio M V Rossi [iD] ✉ & Kelly M McNagny [iD] ✉

## Abstract

Immune responses play an integral role in skeletal muscle regeneration. In the genetically inherited muscle disease Duchenne muscular dystrophy (DMD), muscle regeneration is disrupted, leading to chronic inflammation, fibrosis, and early mortality. Previously, it has been suggested that type-2 innate immune cells, particularly eosinophils and their production of IL-4, play an essential role in effective muscle regeneration after acute injury. We here re-investigate the role of eosinophils in skeletal muscle repair using mice deficient in eosinophils (ΔdblGATA), or deficient in IL-4R/IL-13R signaling through STAT6 (Stat6−/−). We show that neither deficiency has an impact on skeletal muscle regeneration in response to acute injury as quantified by fiber size, immune cell infiltration, or muscle-resident stem cell proliferation. We also investigate the role of STAT6 signaling in mdx:Stat6−/− mice, a model of DMD and, again, find that ablation of STAT6 signaling has no effect on the rate or severity of fibrotic scar formation or disease progression. In contrast to previous models, our data suggest a negligible role for eosinophils and STAT6 signaling in skeletal muscle regeneration after acute or chronic injury.

Keywords Eosinophils; STAT6 Signaling; Skeletal Muscle Regeneration; Duchenne Muscular Dystrophy (DMD)
Subject Categories Immunology; Molecular Biology of Disease; Musculoskeletal System

## Introduction

Skeletal muscle has a remarkable ability to completely regenerate after acute injury without the formation of fibrotic scar tissue (Tidball and Villalta, 2010). This uncanny characteristic of muscle regeneration is mediated by an intricate collaboration between muscle-resident myogenic progenitors (MPs), endothelial cells, pericytes, fibro/adipogenic progenitors (FAPs) that produce trophic factors, and infiltrating immune cells that guide sequential stages of muscle fiber repair (Arnold et al, 2007; Lemos et al, 2015; Juban et al, 2018). Central to this process are both type-1 and type-2 immune cells, and mediators that create the appropriate temporal immune environments for removal of debris and support muscle tissue growth (Sciorati et al, 2016). Strikingly, mice that are immune compromised (or where select immune subsets are absent) show a substantial reduction in the ability to regenerate skeletal muscle (Arnold et al, 2007; Brigitte et al, 2010; Brigitte et al, 2010; Heredia et al, 2013; Babaeijandaghi et al, 2023; Martinez et al, 2010). Conversely, overstimulation of the immune response, as seen in repeated muscle injury, prevents normal muscle regeneration and instead leads to fibrosis and muscle degeneration (Villalta et al, 2015). This is perhaps best exemplified in the case of Duchenne muscular dystrophy (DMD) patients for whom a genetic mutation in an essential structural element of muscle, DYSTROPHIN, leads to continuous destabilization of muscle fiber sarcolemma and repetitive injury (Rodrigues et al, 2016). Unfortunately, efforts to prevent chronic inflammation and the subsequent fibrosis triggered by muscle injury in DMD using standard steroids and non-steroidal anti-inflammatory drugs have shown only limited benefits and, in some cases, enhance pathology (Kourakis et al, 2021). This likely reflects the broad-spectrum effects of these drugs and, potentially, off-target effects on tissue resident muscle cells, or the inhibition of critical immune functions needed to heal tissue injury. Promising new treatments with greater selectivity for specific immune subsets and inflammatory mediators are emerging that target pro-inflammatory, type-1 immune responses (neutrophils, macrophages, TNFα, IFNγ). For example, the kinase inhibitor, nilotinib, blocks fibrotic matrix deposition triggered by TGFβ overexpression in *mdx* mice (Lemos et al, 2015). In addition, depletion of muscle resident macrophages through colony-stimulating factor 1 receptor (CSF1R) blockade alters the composition of muscle fibers in a fashion that renders them damage-resistant in *mdx* mice (Babaeijandaghi et al, 2022). While these are promising avenues, our understanding of the temporal continuum of immune mediators and mechanisms that participate in efficient normal muscle regeneration versus pathologic tissue degeneration remains incomplete and is the principal barrier to effective therapy.

Although all immune responses encompass elements of inflammation and repair, type 1, 2, and 3 immune responses have distinct features (Annunziato et al, 2015). Outside of sterile injury, type-1 responses are primarily mediated by interferon-gamma (IFN-γ) and are associated with defense against intracellular

School of Biomedical Engineering and Department of Medical Genetics, University of British Columbia, Vancouver, BC, Canada. ✉E-mail: fabio@brc.ubc.ca; kelly@brc.ubc.ca

pathogens, involving macrophages, innate lymphoid cells (ILC1s), natural killer (NK) cells, and T helper 1 (Th1) cells; and promoting acute inflammation. Type-3 responses, characterized by IL-17 and IL-22 production from T helper 17 (Th17) cells and innate lymphoid cells (ILC3s), primarily target extracellular bacteria and fungi; and are associated with mucosal barrier integrity and acute inflammatory processes. Type-2 responses are driven by cytokines such as IL-4, IL-5, and IL-13, with a focus on defense against helminths and allergens, involving innate lymphoid cells (ILC2s), T helper 2 (Th2) cells, eosinophils, and basophils; and play a significant role in tissue remodeling and repair (Messing et al, 2020). Type-2 immunity requires STAT6 signaling and mice deficient in STAT6 do not mount a functional type-2 immune response (Kaplan et al, 1996; Akimoto et al, 1998; Zhu et al, 2001; Takeda et al, 1996). This response likely reflects an inherent requirement to repair the damage induced by parasites (Annunziato et al, 2015; Messing et al, 2020). In addition, there is increasing evidence of a homeostatic role for eosinophils in lifelong tissue remodeling (Gurtner et al, 2023). For example, eosinophils are known to play an important role in breast tissue remodeling during post-pregnancy involution and in vaginal tissue remodeling during menstruation (Gouon-Evans et al, 2002; Jeziorska et al, 1995). Recently, this same concept has come to the fore in the field of sterile muscle repair and homeostasis. In a high-profile report from 2013, Heredia et al suggested that eosinophils have a pivotal role in muscle regeneration after acute injury through an eosinophil-to-FAP axis where eosinophil-derived IL-4 guided appropriate FAP cell orchestration of tissue repair (Heredia et al, 2013). Specifically, IL-4 signaling to FAPs was shown to promote muscle regeneration instead of fat deposition and degeneration. More recent studies by us and others, however, have cast doubt on this interpretation. For example, in a recent study we showed that extreme hyper-eosinophilia, induced by over-expression of IL-5, slows muscle regeneration and accelerates fibrosis in *mdx* mice (Theret et al, 2022). Conversely, Sek and colleagues demonstrated that the absence of eosinophils does not impact the development of DMD pathology in four-week-old *mdx* mice, further arguing against a pro-reparative role (Sek et al, 2019).

To more conclusively validate or refute a direct role for eosinophils in muscle repair and gain clarity on the role of IL-4/IL-13 signaling post muscle injury, here we evaluate skeletal muscle regeneration following acute injury in ΔdblGATA and *Stat6−/−* mice which lack eosinophils and transcriptional responses downstream of IL-4R/IL-13R signaling, respectively. We also evaluate adult *mdx:Stat6−/−* mice to study the impact of STAT6 deficiency on chronic muscle injury and fibrosis. Surprisingly, in both scenarios, we find that eosinophils and STAT6 signaling play a negligible role in skeletal muscle regeneration suggesting that targeting these pathways will be of limited therapeutic benefit.

# Results

## Eosinophils are dispensable for muscle regeneration

Previously, it was reported that eosinophil-FAP crosstalk is essential for skeletal muscle regeneration after acute injury (Heredia et al, 2013). To verify this concept as a baseline for our own studies, we assessed the impact of the absence of eosinophils on skeletal muscle regeneration. To

this end, we acutely injured the *tibialis anterior* (TA) muscles of ΔdblGATA mice. The GATA1 transcription factor is essential for earliest stages of eosinophil development and, due to a mutation in the eosinophil-specific GATA-1 promoter, ΔdblGATA mice lack eosinophils from their earliest stages of commitment and development (Yu et al, 2002). We assessed histologically skeletal muscle regeneration at days 7, 14, and 28 post injury (Fig. 1A). Surprisingly, we found an identical ability to repair muscle damage in wild-type (*Wt*) and ΔdblGATA mice: following injury, both body mass and TA mass (normalized to body mass) of ΔdblGATA animals remained unchanged compared to *Wt* mice (Fig. 1B). Similarly, the cross-sectional area (CSA) of regenerative muscle fibers, and the number of nuclei per fiber, was indistinguishable between genotypes (Fig. 1C). Since it was previously suggested that, in the absence of eosinophils, FAPs differentiate into adipocytes and collagen, we assessed adipocyte and collagen deposition but, again, found no discernable difference between ΔdblGATA compared to *Wt* mice (Fig. EV1A). We then assessed the number of eosinophils (SiglecF+ cells) histologically in TA cross sections and, as expected, eosinophils were absent in ΔdblGATA mice at all time points (Fig. 1D). To further characterize the development and resolution of inflammation after injury, we assessed eosinophil and macrophage numbers and polarization in ΔdblGATA and *Wt* mice at 3 days post injury by flow cytometry. This time point coincides with peak eosinophil infiltration as well as transition of pro-inflammatory macrophages (Ly-6C$^{high}$) towards a pro-regenerative phenotype (Ly-6C$^{low}$)[2]. As expected, CD11b$^{low}$ SiglecF+ eosinophils could not be found in ΔdblGATA mice, however, the number and phenotype of Ly6C$^{high}$ and Ly6C$^{low}$ macrophages were identical to *Wt* mice (Fig. 1E) suggesting that macrophage skewing is independent of eosinophils in this tissue. Finally, we assessed MP and FAP cell numbers and proliferation using EdU at day 3 post injury to determine if absence of eosinophils, as previously suggested, impacts muscle resident cells during muscle regeneration. We found that neither MP and FAP numbers, nor their capacity to proliferate was altered in ΔdblGATA mice (Fig. 1F,G). In summary, we conclude that eosinophils are dispensable for macrophage recruitment and skewing, for muscle resident cell proliferation, and for normal muscle regeneration in response to acute skeletal muscle injury.

## STAT6 signaling is dispensable for muscle regeneration

Heredia et al suggested that eosinophils exert their essential function through elaboration of IL-4 and downstream activation of type-2 responsive cells via the IL-4/IL-13 signaling axis during skeletal muscle regeneration. To re-examine these findings, we acutely injured TA muscles of *Stat6−/−* mice. STAT6 is a transcription factor essential for mediating signaling from IL-4/IL-13 receptors such that STAT6-deficient mice are unable to signal via this axis to initiate a functional type-2 immune response (Kaplan et al, 1996; Akimoto et al, 1998; Zhu et al, 2001; Takeda et al, 1996). Again, we assessed the ability to regenerate skeletal muscle in these mice at day 7, 14, and 28 post injury (Fig. 2A). Neither body mass nor TA mass (normalized to body mass) was significantly different between *Stat6−/−* and *Wt* mice (Fig. 2B). Similarly, after injury, the temporal increase in CSA of muscle fibers was identical in both strains as was the number of nuclei per fiber (Fig. 2C). Moreover, comparable to ΔdblGATA mice, adipocyte number and collagen deposition remained equivalent between genotypes (Fig. EV1B). As above, we also histologically assessed the number of eosinophils in TA cross sections and found no impairment in eosinophil infiltration into the muscle in *Stat6−/−* mice (Fig. 2D). Surprisingly, we found slightly more eosinophils after injury in

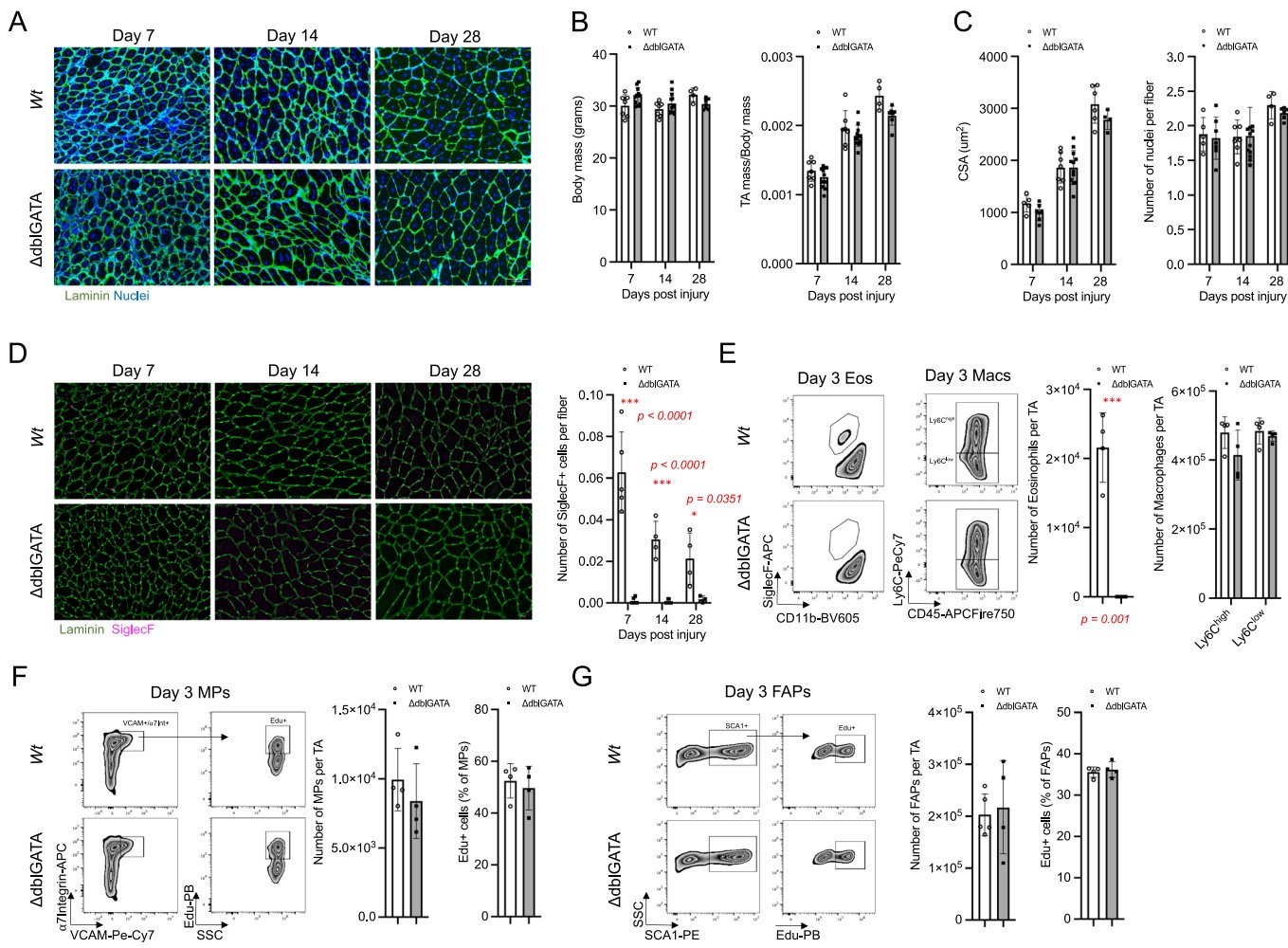

**Figure 1. Eosinophils are dispensable for muscle regeneration.**

*Tibialis anterior* (TA) muscles of wild-type (*Wt*) or ΔdblGATA mice were injured with barium chloride (BaCl$_2$) and harvested to assess muscle regeneration at 7-, 14-, and 28-days post injury (**A–D**). Representative histological laminin and nuclei (DAPI) staining of TA sections (**A**). Mouse body mass and TA mass (normalized to body mass) measurements (**B**). Quantification of cross-sectional area (CSA) and number of nuclei per fiber (**C**). Representative histological SiglecF staining and quantification of SiglecF+ cells in TA muscle sections (**D**). Flow cytometry representative gating and cell number quantification of eosinophils (gated as live/CD45+/CD11b$^{low}$/SiglecF+) and macrophages (gated on live/CD45+/Lin-/Ly6C$^{high/low}$) in TA muscles at 3 days post-BaCl$_2$ injury (**E**). Flow cytometry representative gating, cell number and % proliferation (% EdU+) quantification of myogenic progenitors (MPs) (gated on live/CD45-/CD31-/SCA-1-/α7 integrin+/VCAM+) (**F**). Flow cytometry representative gating, cell number and % proliferation (% EdU+) quantification of fibro/adipogenic progenitors (FAPs) (gated on live/CD45-/CD31-/SCA-1+) (**G**). Number of mice (biological replicates) per group: $n = 4$–9. Two independent replicates were performed for each experiment. *$p < 0.05$; ***$p < 0.001$ by two-tailed, two-sample unequal variance Student's t-test or two-way ANOVA with post hoc testing (Šidak multiple comparison). Scale bars show 100 μm. Error bars represent SD. Source data are available online for this figure.

*Stat6−/−* compared to *Wt* mice 28 days after injury. This may indicate that STAT6-deficiency results in slightly prolonged eosinophil infiltration or delayed eosinophil clearance from skeletal muscle, but without any significant impact on muscle regeneration. We further assessed eosinophil infiltration in *Stat6−/−* mice at 3 days post injury. We found that the number of CD11b$^{low}$SiglecF+ eosinophils, Ly6C$^{high}$ and Ly6C$^{low}$ macrophages were not significantly different between *Stat6−/−* and *Wt* mice (Fig. 2E). We also assessed MP and FAP cell numbers and proliferation rate at day 3 post muscle injury and found that neither MP and FAP numbers nor their capacity to proliferate were altered in *Stat6−/−* compared to *Wt* mice (Fig. 2F,G). Finally, to address whether a partial loss of STAT6 could impact muscle regeneration and hide a potential phenotype, we performed the same

analysis as above on *Wt*, Stat6+/− and *Stat6−/−* mice and we found that neither partial nor complete ablation of STAT6 impact muscle regeneration (Fig. EV2A). In summary, we conclude that STAT6-signaling is dispensable and does not play a significant role in muscle regeneration, inflammatory cell infiltration or muscle resident cell proliferation in response to injury.

## STAT6 signaling is dispensable for repair of chronic muscle injury

Although we failed to identify a role for eosinophils and STAT6 signaling in muscle regeneration after acute injury, the possibility remained that a protective role might only be detectable

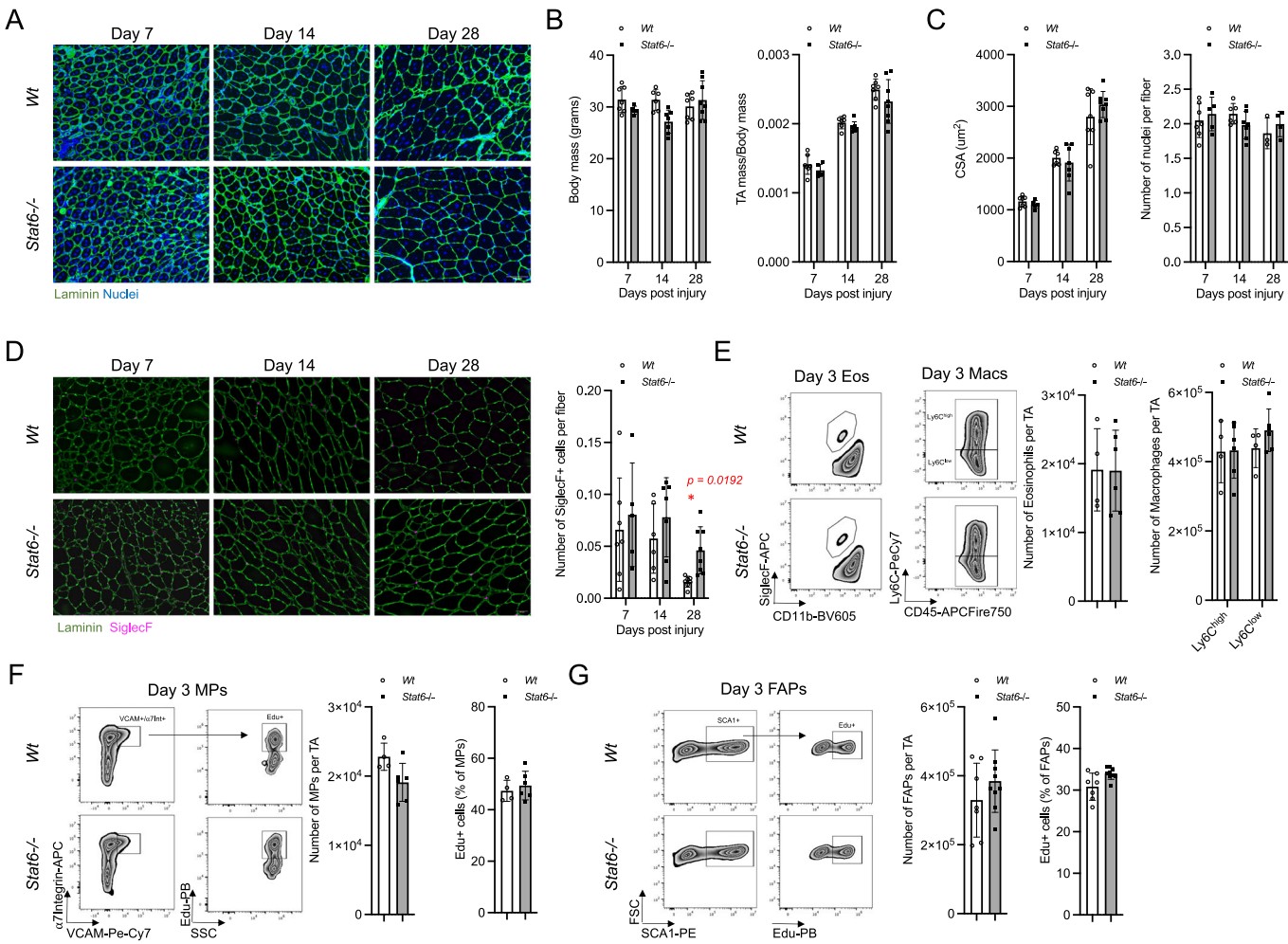

**Figure 2. STAT6 signaling is dispensable for muscle regeneration.**

*Tibialis anterior* (TA) muscles of wild-type (*Wt*) or *Stat6*−/− mice were injured with barium chloride (BaCl₂) and harvested to assess muscle regeneration at 7-, 14-, and 28-days post injury (A–D). Representative histological laminin and nuclei (DAPI) staining of TA sections. Mouse body mass and TA mass (normalized to body mass) measurements (B). Quantification of cross-sectional area (CSA) and number of nuclei per fiber (C). Representative histological SiglecF staining and quantification of SiglecF+ cells in TA muscle sections (D). Flow cytometry representative gating and cell number quantification of eosinophils (gated as live/CD45+/CD11b$^{low}$/SiglecF+) and macrophages (gated on live/CD45+/Lin-/Ly6C$^{high/low}$) in TA muscles at 3 days post-BaCl₂ injury (E). Flow cytometry representative gating, cell number and % proliferation (% EdU+) quantification of myogenic progenitors (MPs) (gated on live/CD45-/CD31-/SCA-1-/α7 integrin+/VCAM+) (F). Flow cytometry representative gating, cell number and % proliferation (% EdU+) quantification of fibro/adipogenic progenitors (FAPs) (gated on live/CD45-/CD31-/SCA-1-) (G). Number of mice (biological replicates) per group: n = 4–8. Two independent replicates were performed for each experiment. *p < 0.05 by two-way ANOVA with post hoc testing (Šidak multiple comparison). Scale bars show 100 μm. Error bars represent SD. Source data are available online for this figure.

in response to more lengthy chronic damage or repetitive injury. To evaluate this possibility, we assessed the effects of STAT6-deficiency in adult (12-week-old) *mdx* mice. We began by assessing immune cell infiltration (eosinophils and macrophages) as well as muscle resident cell (MP and FAP) number and proliferation in pooled hindlimb muscles (TA, *gastrocnemius* (Gastroc), and *quadricep* (Quad)). We found no differences between *mdx:Stat6*−/− compared to *mdx:Stat6*+/− mice in terms of the number of infiltrating immune cells or the number of muscle resident cells (Fig. 3A). MPs were equally proliferative (~15% EdU+) in *mdx:Stat6*−/− compared to *mdx:Stat6*+/− mice while proliferating FAPs could not be found in either group. Importantly, when comparing *mdx:Stat6*+/+ with *mdx:Stat6*+/−, and *mdx:Stat6*−/− mice, muscle repair was identical, indicating that partial loss of STAT6

does not confound our findings (Fig. EV2B). Typically, hindlimb muscles show acute lesions in young (~4-week-old) *mdx* mice while in adult (8–20-week-old) *mdx* mice, muscles regenerate with little fibrosis (Bulfield et al, 1984). To accelerate chronic injury in adult *mdx* mice, we manually micro-damaged TA muscles of *mdx:Stat6*−/− using a well-documented needle microdamage approach (Desguerre et al, 2012). We assessed the histology of TA sections from *mdx:Stat6*−/− and *mdx:Stat6*+/− mice with and without micro-damage (Fig. 3B–E). While we observed elevated numbers of SiglecF+ cells in micro-damaged (MD) TA compared to undamaged (No MD) TA, we found no distinguishable differences between the numbers of these cells in *mdx:Stat6*−/− compared to *mdx:Stat6*+/− mice (Fig. 3B). Further, when comparing MD TA compared to No MD TA, we observed

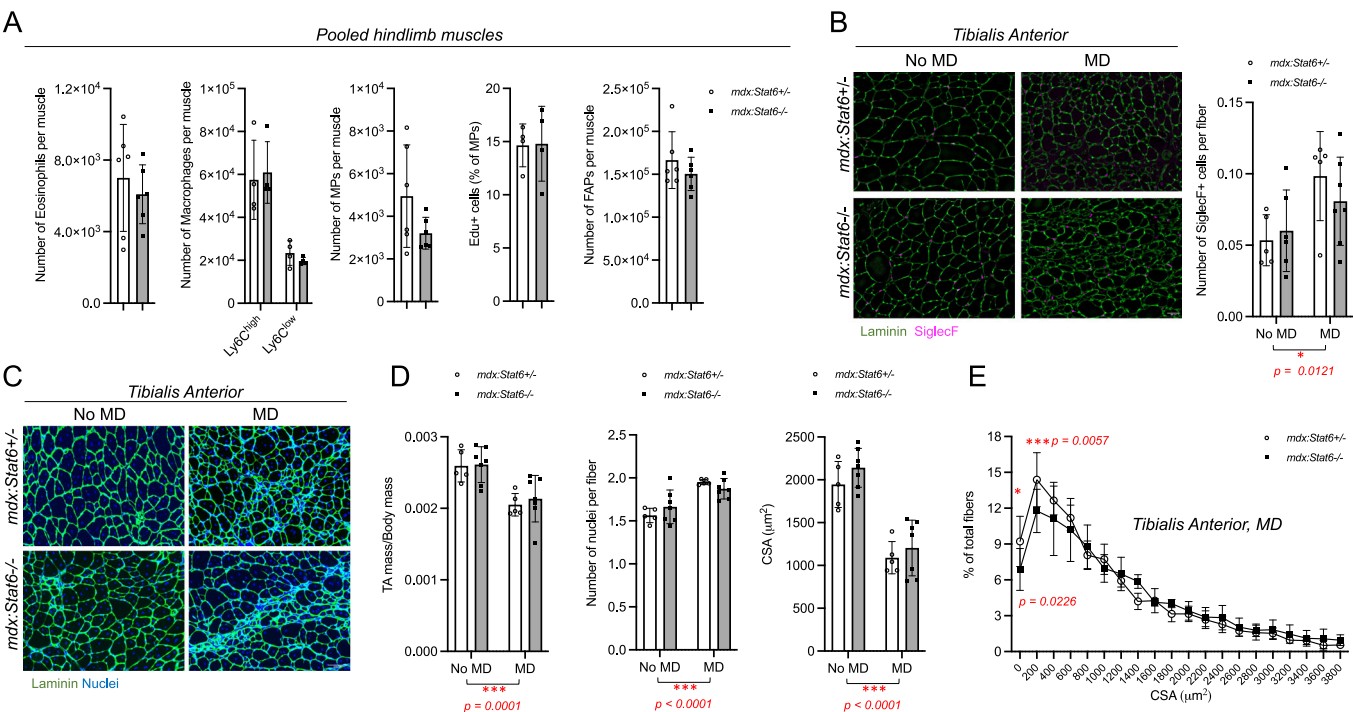

**Figure 3.  STAT6 signaling and eosinophils are dispensable for the repair of chronic muscle injury.**

Flow cytometry cell number quantification of eosinophils (gated as live/CD45+/CD11b^low/SiglecF+) and macrophages (gated on live/CD45+/Lin-/Ly6C^high/low), cell number and % proliferation (% EdU+) quantification of myogenic progenitors (MPs) (gated on live/CD45-/CD31-/SCA-1-/α7 integrin+/VCAM+), and cell number quantification of fibro/adipogenic progenitors (FAPs) (gated on live/CD45-/CD31-/SCA-1+) in pooled undamaged *tibialis anterior* TA, *gastrocnemius* (Gastroc) and *quadricep* (Quad) muscles (A). At age 3 months, the right TA of *mdx:Stat6+/−* and *mdx:Stat6−/−* mice were micro-damaged (MD) while the left TA was not damaged (No MD); Tissues were collected at 14 weeks of age (B–E). Representative histological laminin and SiglecF staining and quantification of SiglecF+ cells in TA muscle sections (B). Representative histological laminin and nuclei (DAPI) staining of TA muscle sections (C). TA mass (normalized to body mass) measurements, quantification of number of nuclei per fiber, cross-sectional area (CSA) and binned CSA (D, E). Number of mice (biological replicates) per group: n = 4–7. Two independent replicates were performed for each experiment. *p < 0.05; **p < 0.001 by two-way ANOVA with post hoc testing (Šidak multiple comparison). Scale bar represents 100 μm. Error bars represent SD. Source data are available online for this figure.

significantly reduced muscle mass (normalized to body mass), as well as a significantly higher number of nuclei per fiber and significantly smaller CSA (Fig. 3C,D) in both *mdx:Stat6+/−* and *mdx:Stat6−/−*. However, when compared against each other, these changes remained indistinguishable across genotypes, suggesting that STAT6 signaling is dispensable for muscle repair in this chronic muscle injury model. Interestingly, a binned CSA analysis showed a small but significant reduction in the number of small fibers (CSA = 0–400 μm) in MD TAs of *mdx:Stat6−/−* compared to *mdx:Stat6+/−* (Fig. 3E). While small, and likely inconsequential for long term muscle regeneration or degeneration, it suggests that the lack of STAT6 is not only dispensable in this model but may be slightly beneficial. That aside, both the deposition of adipocytes and collagen also remained indistinguishable in MD TAs of *mdx:Stat6−/−* compared to *mdx:Stat6+/−* (Fig. EV1C).

## Eosinophils are dispensable for repair of chronic muscle injury

To assess the impact of eosinophils on chronic muscle injury directly, we used an anti-IL-5 neutralizing antibody (α-IL-5) to deplete eosinophils in *mdx* mice prior to performing TA

microdamage. To confirm eosinophil depletion, we assessed the frequency of circulating eosinophils and found a significant reduction at 7- and 14-days post α-IL-5 treatment compared to *mdx* mice treated with an isotype control (Fig. 4A). In addition, at the time of TA tissue collection, we also found a significant reduction in SiglecF+ cells in both MD TA and No MD TA sections of α-IL-5 treated *mdx* mice (Fig. 4A). However, TA mass (normalized to body mass) and the number of nuclei per fiber were not significantly different between α-IL-5 and isotype-treated *mdx* mice (Fig. 4B). While we did find significantly reduced CSA of muscle fibers in MD TAs compared to No MD TAs, there was also no difference in mean CSA when comparing α-IL-5 and control treatment groups (Fig. 4B). Intriguingly, we again observed a small but significant reduction in the number of small fibers in α-IL-5-treated *mdx* mice which suggests that a reduction of eosinophils, similar to the reduction of STAT6 signaling, is actually slightly beneficial rather than detrimental to muscle regeneration in this chronic injury model (Fig. 4C). Overall, we conclude that eosinophils and STAT6 signaling are largely expendable during chronic injury of hindlimb muscles in 12-week-old *mdx* mice. To further confirm this result, we next assessed the long-term impact of STAT6 signaling on chronic muscle injury.

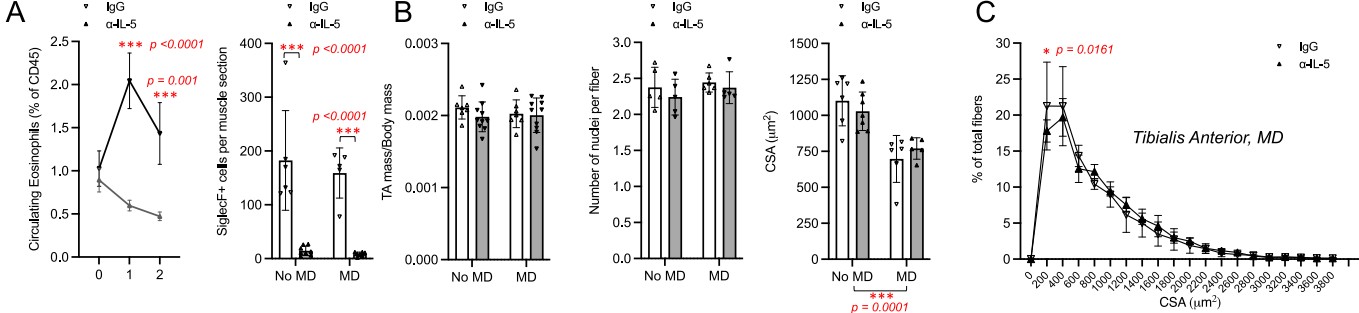

**Figure 4. Eosinophils are dispensable for the repair of chronic muscle injury.**

Quantification of circulating eosinophil frequency and SiglecF+ cells per MD and No MD TA sections of αIL-5 or IgG intraperitoneally injected mdx mice (A). TA mass (normalized to body mass) measurements, quantification of number of nuclei per fiber, cross-sectional area (CSA) and binned CSA (B, C). Number of mice (biological replicates) per group: $n$ = 4–7. Two independent replicates were performed for each experiment. *$p < 0.05$; ***$p < 0.001$ by two-way ANOVA with post hoc testing (Šidak multiple comparison). Scale bar represents 100 μm. Error bars represent SD. Source data are available online for this figure.

## STAT6 signaling does not contribute to DMD pathology

Fibrosis and subsequent functional impairment of skeletal muscles are hallmark features of DMD and, in both human and *mdx* mice, progressively worsen with age. To evaluate whether STAT6 signaling impacts the longer term DMD pathology with advanced age, we analyzed the phenotype and functional properties of hindlimb skeletal muscles in 3- and 10-month-old *mdx:Stat6−/−* mice. Of interest, we found that the Gastroc was the only hind muscle that showed significantly lower muscle mass at 10 months compared to 3 months of age which suggests an accelerated pathology in this muscle compared to the other hind limb muscle types (Fig. 5A,B).

That aside, we again found that neither body mass nor muscle mass (normalized to body mass) was significantly different at 3 or 10 months of age in *mdx:Stat6−/−* compared to *mdx:Stat6+/−* mice (Fig. 5A,B). We further assessed if, functionally, *mdx:Stat6−/−* differed from *mdx:Stat6+/−* mice. To this end, we measured the grip strength (normalized to body mass) of the front legs in isolation, or the front and back legs combined (Fig. 5C). Although we found that age did not significantly impact the grip strength of the front legs alone, we found the combined force of front and back legs showed a significant decline with age, likely reflecting our observation of a decline in Gastroc muscle mass. But, once again, the grip strength of *mdx:Stat6−/−* mice compared to *mdx:Stat6+/−* mice remained indistinguishable at 3 and 10 months of age.

We then assessed the level of skeletal muscle fibrosis as well as the level of inflammation in leg muscles of adult and aged *mdx:Stat6−/−* compared to *mdx:Stat6+/−* mice (Fig. 5D–F). Muscles were stained with Picrosirius red (PSR) and Hematoxylin and Eosin (H&E) to visualize and quantify collagen deposits and inflammation, respectively. None of the hindlimb muscles showed significant evidence of collagen deposition at 3 months of age and, matching our findings regarding muscle mass, only the Gastroc showed a significant increase in collagen deposition at 10 months of age. In addition, collagen deposition was not significantly changed in hindlimb muscles of adult or aged *mdx:Stat6−/−* compared to *mdx:Stat6+/−* mice. We also found that inflammation was unchanged in TA muscles of adult and aged *mdx* mice while both Gastroc and Quad showed slightly lower levels of inflammation with age. Again, no differences were observed

in terms of inflammation when comparing *mdx:Stat6−/−* and *mdx:Stat6+/−* mice.

Because the DMD phenotype of *mdx* mouse hindlimb muscles was still relatively mild both at 3 and 10 months of age, we also assessed fibrosis development and inflammation in the diaphragm, since this muscle is well-known to exhibit DMD-pathology much earlier than hindlimb muscles due to the repeated, continual, heavy use in normal respiration (Stedman et al, 1991). Indeed, we observed considerable collagen deposition in the diaphragm at 3 months of age and this continued to increase further with age (Fig. 5G–I). Despite the earlier onset of pathology in this muscle, once again we saw no differences in collagen deposition in diaphragms of *mdx:Stat6−/−* compared to *mdx:Stat6+/−* mice at either 3 or 10 months (Fig. 5G,H). Likewise, we saw no differences in the level of inflammation when comparing these mouse strains: although inflammation was greater in aged mice compared to younger adults, there were no differences when comparing the two genotypes (Fig. 5G,I). Finally, adipocyte deposition remained similar in the diaphragm of 10-month-old *mdx:Stat6−/−* compared to *mdx:Stat6+/−* mice (Fig. EV1D). In summary, after exhaustive analyses, we failed to detect any evidence for a role of STAT6 signaling in modulating disease onset or outcome in *mdx* mice.

## Muscle regeneration differs in BALB/c and C57BL/6 mice

Interestingly, while our studies have been performed with mice on a C57BL/6 background, Heredia et al performed selected experiments on mice from a BALB/c background. BALB/c and C57BL/6 background strains have been shown to differ in terms of their immune response, which has been well-documented in various contexts (Fukushima et al, 2006; Plum et al, 2023; Busch et al, 2016; Pae et al, 2010; Walsh et al, 2008; Humbles et al, 2004; Lee et al, 2004; Wills-Karp and Karp, 2004). This prompted us to test if regeneration after acute muscle injury showed any profound differences between these two strains that could explain such differences between our results. To this end, we acutely injured the TA muscles of *Wt* BALB/c and *Wt* C57BL/6 mice with BaCl2 and

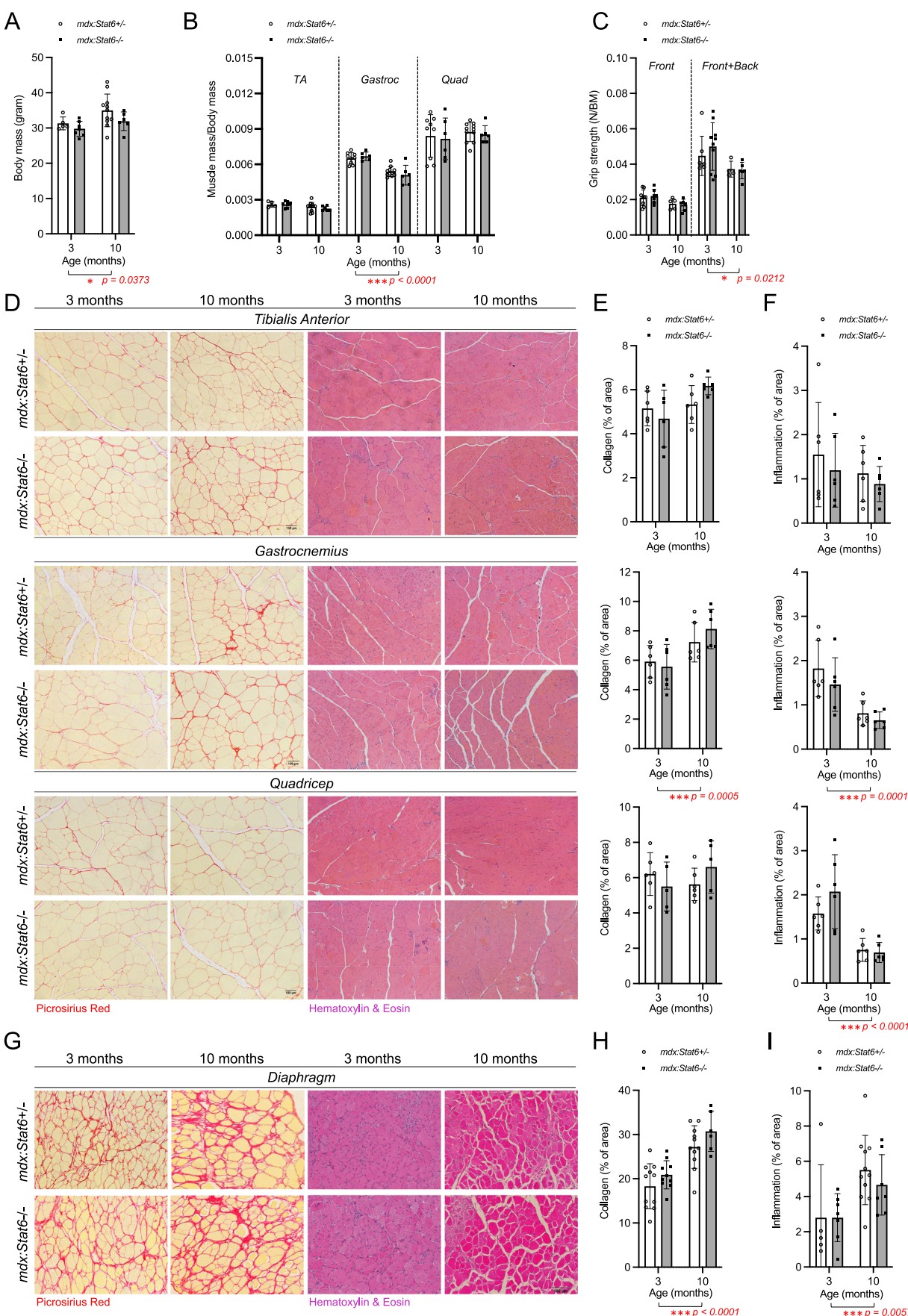

**Figure 5. STAT6 signaling does not contribute to DMD pathology.**

At age 3 months and 10 months, mouse body mass (**A**), muscle mass (normalized to body mass) of *tibialis anterior* (TA), *gastrocnemius* (Gastroc) and *quadricep* (Quad), and grip strength force (normalized to body mass) were measured in *mdx:Stat6+/−* and *mdx:Stat6−/−* mice (**C**). Representative histological picrosirius red (PSR) and hematoxylin and eosin (H&E) staining of TA (top), Gastroc (middle), and Quad (bottom) at 3 months and 10 months (**D**). Quantification of % collagen based on PSR staining (**E**) and inflammation based on H&E staining (**F**) of TA, Gastroc and Quad at age 3 months and 10 months. Representative histological PSR and H&E of diaphragm (**G**) and % collagen and inflammation quantification (**H, I**). Number of mice (biological replicates) per group: $n = 6–11$. Two independent replicates were performed for each experiment. *$p < 0.05$; ***$p < 0.001$ by two-way ANOVA with post hoc testing (Šidak multiple comparison). Scale bar represents 100 µm. Error bars represent SD. Source data are available online for this figure.

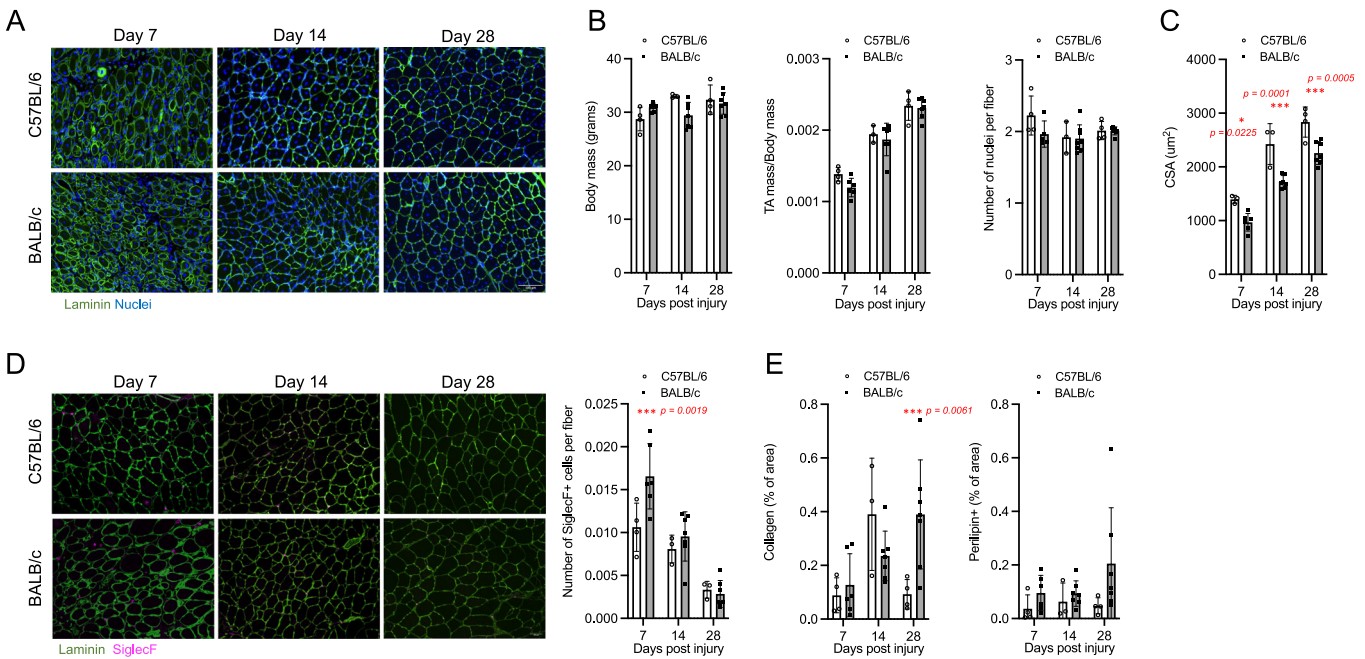

**Figure 6. Muscle regeneration differs in BALB/c and C57BL/6 mice.**

*Tibialis anterior* (TA) muscles of C57BL/6 and BALB/c mice were injured with barium chloride (BaCl₂) and body mass, TA mass (normalized to body mass), number of nuclei per fiber, and cross-sectional area (CSA) were quantified to assess muscle regeneration at 7-, 14-, and 28-days post injury (**A–C**). To assess eosinophil infiltration, collagen and adipocyte deposition, number of SiglecF+ cells, collagen (%of area), and perilipin (% of area) were quantified respectively (**D, E**). Number of mice per group: $n = 3–7$. Two independent replicates were performed for each experiment. *$p < 0.05$; ***$p < 0.001$ by two-way ANOVA with post hoc testing (Šidak multiple comparison). Error bars represent SD.

histologically assessed skeletal muscle regeneration at days 7, 14, and 28 post injury (Fig. 6A). Following injury, both body mass, TA mass (normalized to body mass) and number of nuclei per fiber of BALB/c animals remained similar to C57BL/6 mice (Fig. 6B). However, when assessing the CSA of regenerative muscle fibers, we found that BALB/c mice displayed smaller muscle fibers at all timepoints, indicating a different kinetic of muscle regeneration (Fig. 6C). We assessed the number of eosinophils (SiglecF+ cells) histologically in TA cross sections and, while not significant at day 14 or day 28 post injury, we found significantly elevated numbers of SiglecF+ cells in BALB/c mice compared to C57BL/6 mice at day 7 post injury (Fig. 6D). Finally, we also observed significantly more collagen deposition at day 28 post injury and a trend towards an increase in adipocytes in BALB/c animals at all timepoints after injury (Fig. 6E). In conclusion, the above results indicate that, compared to C57BL/6, BALB/c mice have a tendency towards delayed muscle regeneration and increased eosinophil infiltration

early during regeneration. These differences are more profound than any differences observed when comparing loss of eosinophils or STAT6 deficiency on a uniform C57BL/6 background.

## Discussion

While type-1, type-2, and type-3 immune responses play pivotal roles in the protection from viral, parasitic, and bacterial infections, respectively, the role of these responses in tissue homeostasis and sterile injury, particularly in non-barrier organs, remains far more controversial. In this study, we explored the importance of STAT6-signaling and eosinophils (typically associated with Th2 inflammation) in the context of skeletal muscle regeneration. We found that neither the absence of eosinophils nor impaired STAT6 signaling impacts skeletal muscle regeneration after acute injury. This runs counter to a cardinal study that previously investigated the role of

both eosinophils and their signaling via the IL-4 axis (upstream of STAT6) in the context of acute skeletal muscle injury. Previously, Heredia et al suggested that eosinophils are the primary source of IL-4 release during muscle regeneration after acute injury and that eosinophil-FAP crosstalk via the IL-4/IL-13 signaling axis is required for muscle regeneration. Specifically, they suggested that eosinophils promote FAP proliferation and prevent muscle fatty degeneration (Heredia et al, 2013). Here, using two separate models, we show that, in the absence of eosinophils or functional STAT6 signaling, skeletal muscle regeneration is entirely unaltered compared to wild-type mice by all major measurements of regeneration. These include muscle fiber CSA, the number of nuclei per fiber, inflammatory cell infiltration and muscle resident cell proliferation. Instead, we found that eosinophils are completely dispensable for FAP proliferation using age- and sex-matched controls. Thus, our findings challenge the previously attributed notion that these aspects of Th2 immune response are pivotal in repair of acute skeletal muscle injury (Heredia et al, 2013).

We further showed that impaired STAT6 signaling also does not impact DMD pathology of adult and aged *mdx* mice and note that several previous studies have also investigated the role of eosinophils, in this context (Theret et al, 2022; Sek et al, 2019; Wehling-Henricks et al, 2008; Kastenschmidt et al, 2021; Cai et al, 2000; Jiang et al, 2023). For example, a previous study showed that elevated eosinophil counts in muscle of DMD patients correlated with disease progression (Jiang et al, 2023). Initially, it was suggested that eosinophils contribute to development of skeletal muscle fibrosis in aged mice through the release of major basic protein 1 (MBP-1) and that eosinophils mediate muscle fiber lysis in 4-week-old *mdx* mice (Wehling-Henricks et al, 2008). In contrast, a more recent study demonstrated that 4-week-old eosinophil-deficient PHIL (EPO-DTA; JAX#036945) and ΔdblGATA *mdx* mice showed no significant impact on acute muscle pathology (Sek et al, 2019). This latter study additionally showed that hyper-eosinophilia in 4-week-old IL-5 transgenic (CD3d-Il5; JAX#036943) *mdx* mice did not contribute significantly to muscle pathology. Conversely, we recently showed that adult (3 months+) and aged IL-5 transgenic *mdx* mice present with accelerated DMD pathology in terms of fibrotic matrix deposition arguing for a pathogenic rather than a beneficial role (Theret et al, 2022). Our work shows that eosinophil depletion in chronic skeletal muscle injury of adult *mdx* mice has no major impact on muscle regeneration or disease progression. It is important to note that although the presence of eosinophils in muscle lesions of both DMD patients and *mdx* mice is well-documented, the study of these cells in muscle pathology has previously led to inconsistent or even contradictory results (Kastenschmidt et al, 2021; Jiang et al, 2023).

It is evident that the age at which *mdx* mice are studied can significantly impact interpretation since, at 4 weeks of age, *mdx* mice undergo bouts of acute pathology and inflammation followed by efficient regeneration while adult mice start to show fibrosis deposition that progresses with age (Villalta et al, 2015; Stedman et al, 1991). Our results indicate, unequivocally, that STAT6 signaling is not a major contributor to progression or resolution of muscle degeneration in the DMD model and this also matches our observation that neither this signaling axis nor eosinophils are significantly involved in skeletal muscle regeneration. Given these observations, we hypothesize that eosinophils are unlikely to contribute, in a significant manner, to the fibrotic stage of the disease, at least not through STAT6 signaling.

Importantly, we note that Heredia et al used mice on a mixture of genetic backgrounds including the type-2-skewed BALB/c mouse strain, while all mice used in the current study were on the type-1-biased C57BL/6 background. It is well-documented that these two background strains can differ in terms of their immune response in various contexts (Fukushima et al, 2006; Plum et al, 2023; Busch et al, 2016; Pae et al, 2010; Walsh et al, 2008; Humbles et al, 2004; Lee et al, 2004; Wills-Karp and Karp, 2004). For example, compared to BALB/c mice, C57BL/6 mice show heightened airway responses following mucosal injury (Fukushima et al, 2006). In contrast, in the context of chronic allergen challenge, only BALB/c mice and not C57BL/6 mice show mast cell accumulation (Plum et al, 2023). Perhaps more tellingly, it is well-known that eosinophil deficiency in these two strains show striking differences in the pathology linked to type-2-driven airway inflammation with C57BL/6 mice being protected from airway hyper-responsiveness (AHR) in the absence of eosinophils while BALB/c mice develop AHR even in the complete absence of eosinophils (Humbles et al, 2004; Lee et al, 2004; Wills-Karp and Karp, 2004). Indeed, there are numerous additional studies highlighting profoundly different type-2-skewed immune responses of BALB/c and C57BL/6 strains (Fukushima et al, 2006; Plum et al, 2023; Busch et al, 2016; Pae et al, 2010; Walsh et al, 2008; Humbles et al, 2004; Lee et al, 2004; Wills-Karp and Karp, 2004). When comparing muscle injury and regeneration between both strains, we indeed noted differences including significantly increased early eosinophil infiltration, delayed regeneration, more collagen deposition, and a trend towards increased adipocyte deposition in BALB/c compared to C57BL/6 mice. Importantly, these differences were more significant than those observed when comparing eosinophil- or STAT6-deficient animals on the C57BL/6 background. In addition, it is likely that these differences in BALB/c responses may be exacerbated when BALB/c mice also lack eosinophils or eosinophil-mediated signaling and these observations lend further support to the idea that strain differences are indeed the reason for widely different results with regards to the importance of eosinophils and STAT6 signaling in muscle regeneration. Notably, this underscores the likelihood that utilizing strains with differing genetic backgrounds within the same study contributes significantly to conflicting and confounding conclusions. In summary, we hypothesize that wider inferences on the significance of eosinophils and type-2 immune signaling derived from studies performed on a BALB/c genetic background need to be interpreted with significant caution as subtle differences linked to this hyper-responsive background may not be broadly extrapolatable to other mouse strains or more generally to human disease. We do note that due to the length of time required to perform the reported muscle damage models and the limited availability of identical genetic lesions in type-2 immunity on multiple genetic backgrounds, we did not perform a complete comparison of the BALB/c and C57BL/6 genetic background effects on acute and chronic muscle injury but feel this is warranted in the future.

Importantly, this research has implications for future studies that aim to pursue the development of targeted immunotherapeutics for DMD. Other targeted immunomodulatory treatments and particularly those of type-1 innate immune cells and mediators, have shown significant promise for the treatment of DMD (Lemos et al, 2015; Gouon-Evans et al, 2002). This avenue of research is of great importance since inflammation is a major driver of disease development, yet, broad spectrum anti-inflammatory drugs show little effect, likely due to the essential contributions of immune cells

to muscle repair (Kourakis et al, 2021). To this end, identifying the appropriate target is a critical step towards a successful treatment and our results suggest, in contrast to previous research, eosinophils and STAT6 signaling are not likely to be relevant targets for the development of efficacious immunotherapeutics.

# Methods

### Reagents and tools table

| Reagent/Resource | Reference or Source | Identifier or Catalog Number |
|---|---|---|
| **Experimental models** | | |
| C57BL/6J | Jackson Laboratory | #000664 |
| C.129S1(B6)-*Gata1^{tm6Sho}*/J | Jackson Laboratory | #005653 |
| C.129S2-*Stat6^{tm1Gru}*/J | Jackson Laboratory | #005977 |
| C57BL/10ScSn-*Dmd^{mdx}*/J | Jackson Laboratory | #001801 |
| BALB/C.CJ | Jackson Laboratory | #000651 |
| **Antibodies** | | |
| Ly-6G (Gr-1) | BD Bioscience | 1A8 |
| NK1.1 | Biolegend | PK136 |
| CD3e | Biolegend | 145-2C11 |
| SiglecF | Biolegend | E50-2440 |
| CD11b | BD Bioscience | M1/70 |
| Ly-6C | Biolegend | HK1.4 |
| CD45 (Pan) (Muscle panel) | Biolegend | 30-F11 |
| CD45 (Pan) (Eosinophil panel) | AbLab | I3/2 |
| CD31 | Ebioscience | 390 |
| A7integrin | Ablab | R2F2 |
| VCAM | Biolegend | 429 (MVCAM.A) |
| SCA-1 | Ebioscience | D7 |
| Anit-IL-5 blocking antibody | Invitrogen | TRFK5 |
| Anti-Laminin | Abcam | #ab11575 |
| Anti-SiglecF | Invitrogen | #14-1702-80 |
| Anti-Perilipin | Sigma | #P1873 |
| Anti-Collagen | SouthernBiotech | #1310-01 |
| Secondary #1 | Invitrogen | #A21247 |
| Secondary #2 | Invitrogen | #A11008 |
| Secondary #3 | Invitrogen | #A21447 |
| Secondary #4 | Invitrogen | #A21206 |
| 4′,6-diamidino-2-phenylindole (DAPI) | Invitrogen | #D3571 |
| **Chemicals, Enzymes and other reagents** | | |
| Click-iT Plus EdU kit | Thermo Fisher Scientific | #C10636 |
| Barium chloride | Sigma | #B0750 |
| 2,2,2-Tribromoethanol | Sigma | #T48402 |
| Collagenase D | Millipore Sigma | #11 088 882 001 |
| Dispase II | Millipore Sigma | #04 942 078 001 |

| Reagent/Resource | Reference or Source | Identifier or Catalog Number |
|---|---|---|
| Fixable viability dye | Thermo Fisher Scientific | #L34980 |
| **Software** | | |
| FlowJoTM | | |
| GraphPad Prism | | |
| QuPath v.0.4.3 | | |
| Fiji ImageJ | | |
| **Other** | | |
| Digital force gauge | Chatillon | |
| LSRII | BD Aria | |
| Eclipse Ni microscope | Nikon | |
| Cytoflex LX | Beckman | |

## Mice

C57BL/6J (#000664), C.129S1(B6)-*Gata1^{tm6Sho}*/J (ΔdblGATA, #005653), C.129S2-*Stat6^{tm1Gru}*/J (*Stat6^{-/-}*, #005977), C57BL/10ScSn-*Dmd^{mdx}*/J (*mdx*, #001801), BALB/C.CJ (#000651) strains were purchased from Jackson Laboratory and maintained at the UBC Biomedical Research Centre animal facility. Mice were maintained in an enclosed and pathogen-free facility, housed in standard cages under 12 h light/dark cycles and fed ad libitum with a standard chow diet. All experimental procedures were approved by the UBC Animal Care Committee. Mice were age-matched, and all experiments were conducted on male mice. To assess cell proliferation, 0.5 mg of EdU were injected intraperitoneally the evening (14 h) before tissue harvest (Click-iT Plus EdU kit; Thermo Fisher Scientific, #C10636).

## Muscle injury

Acute muscle injury was induced with 40 μl intramuscular injections of 0.9% (v/v) barium chloride (BaCl₂, Sigma-Aldrich #B0750) in the *tibialis anterior* (TA) muscle. To enhance chronic injury, TA muscles of *mdx* mice were manually micro-damaged with micro-needle pricks (15x/day for 15 consecutive days) as previously described (Stedman et al, 1991).

## Eosinophil depletion

To deplete eosinophils, *mdx* mice were intraperitoneally injected with an anti-IL-5 blocking antibody (Invitrogen, clone TRFK5, 1 mg/kg) at the beginning as well as 1 week into the TA microdamage protocol (Garlisi et al, 1999). Control animals were injected with 1 mg/kg of rat IgG isotype control (Ablab). Blood was taken prior to the first injection, at 1 week prior to the second injection and at time of tissue collection to confirm eosinophil depletion by flow cytometry. Blood was harvested from the lateral saphenous vein. 2–3 drops of blood were harvested in 1X PBS 2 mM EDTA. After centrifugation, red blood cells were hemolyzed using ACK buffer (Gibco). After two washes with FACS buffer (1X PBS 2 mM EDTA 2% FBS), white blood cells were incubated with

Fc Block (AbLab) in FACS Buffer and then stained for eosinophils (Reagents and Tools Table). Finally, samples were then acquired on a LSRII (BD Aria), and data was analyzed using FlowJoTM (BD Biosciences). Eosinophils are CD45+/Lin-/CD11b$^{low}$/SiglecF+.

## Grip strength testing

Grip strength testing was performed using a digital force gauge (Chatillon) attached to a grid. After recording the mouse weight for normalization, the mouse is lowered over the grid keeping the torso horizontal and allowing only its forepaws to attach to the grid before any measurements are taken. The mouse is gently pulled by its tail ensuring the mouse grips the top portion of the grid and the torso remains horizontal. The maximal grip strength value of the mouse is recorded. This procedure is repeated twice more to obtain 3 forelimb grip strength measurements, with 1 min of rest in between each set. The day after (to let the mice recover from the effort), the same procedure is done with both forelimb and hindlimb on the grid.

## Histology and imaging

For Picrosirius red (PSR) and Hematoxylin and Eosin (H&E) staining, diaphragm muscles were harvested and fixed in 1% paraformaldehyde (PFA) for 24 h prior to transfer to 70% ethanol. Following paraffin embedding, muscles were sectioned at a thickness of 5 μm. PSR and H&E staining were performed according to standard protocols (Wax-it Histological Services Inc.). For immunostaining with laminin and SiglecF, TA muscles were harvested, snap frozen in liquid nitrogen precooled isopentane and stored at −80 °C. Tissues were sectioned at a thickness of 10 μm on a Leica Cryostat. Sections were fixed in 4% PFA for 10 min at room temperature (RT), rinsed with 1X PBS and incubated in blocking buffer (1X PBS, 3% serum, 0.3% Triton X-100) prior to staining with primary antibodies (Anti-Laminin: Abcam #ab11575, Anti-SiglecF: Invitrogen #14-1702-80, Anti-Perilipin: Sigma #P1873, Anti-Collagen: SouthernBiotech #1310-01) at 4 °C, overnight. The following day, sections were washed 3 times with 1X PBS before incubation for two hours at RT with secondary antibodies (Invitrogen #A21247, Invitrogen #A11008, Invitrogen #A21447, Invitrogen #A21206) followed by staining of nuclei with 0.6 mM of 4′,6-diamidino-2-phenylindole (DAPI) for 10 min in the dark (Invitrogen #D3571). Brightfield and fluorescent images were captured at 10X and 20X objectives with a Nikon Eclipse Ni microscope with a dual-use monochromatic and color imaging camera (Nikon Digital Sight DS-U3 for brightfield, Qimaging Retiga EXi for fluorescence), operated via the NISElements software. The cross-sectional area (CSA) was automatically segmented with the Cellpose 2.0 human-in-the-loop training model (Stringer et al, 2021; Pachitariu and Stringer, 2022). Selection and quantification of damaged muscle fiber CSA was completed on QuPath v.0.4.3 (Bankhead et al, 2017). Centrally nucleated fibers (laminin and DAPI stained) were quantified with Fiji (ImageJ, version 2.0.0-rc/69/1.52n, NIH, MD). For fiber size analyses, a minimum of 500 nucleated fibers were quantified from acutely injured muscle and a whole TA section was quantified from chronically injured muscle (mdx mice). For SiglecF+ cell number quantifications, pictures were taken at 20X, 6 pictures/mouse were quantified, and the mean was used as the final result. Fiji was used to quantify PSR staining as described previously (Lo et al, 2019). The quantification of inflammation in H&E images was also performed with Fiji and the protocol for quantification

was as described previously and adapted for muscle (Hughes and McNagny, 2016). Briefly, a complete muscle section was imaged and manually divided into 8–10 grids. In each grid, areas of inflammation were manually measured. All areas of inflammation were added up and taken as a percent of total muscle area. All quantifications were conducted in a blinded manner.

## Flow cytometry

Mice were anesthetized with Avertin (2,2,2-Tribromoethanol, Sigma-Aldrich #T48402) and perfused with 20 ml of 1X PBS, 2 mM EDTA. Muscle tissues were harvested, facias were removed, and muscles were manually minced followed by digestion with 1.5 U/ml collagenase D (Millipore Sigma; #11 088 882 001), 2.4 U/ml Dispase II (Millipore Sigma; #04 942 078 001) and 10 mM CaCl$_2$ at 37 °C for one hour. Cell suspensions were diluted with FACS Buffer (1X PBS, 2% FBS, 2 mM EDTA) followed by filtration with a 40 μm filter. Isolated cells from muscle tissues were incubated for 20 min at 4 °C in FACS buffer containing Fc Receptor-blocking antibody (AbLab) and fixable viability dye (ThermoFisher; #L34980) and stained for 30 min at 4 °C with FACS buffer containing antibody cocktails (Reagents and Tools Table). To assess proliferation, EdU incorporation was stained according to manufacturer instructions (Click-iT Plus EdU kit; Thermo Fisher Scientific, #C10636). Data acquisition was performed with Cytoflex LX (Beckman), and data analysis was performed using the FlowJoTM software (BD Biosciences). Representative gating strategies are shown in Fig. EV3.

## Statistical analysis

Sample size and statistical tests are indicated in figure legends. All graphs and statistical tests were generated using GraphPad Prism (GraphPad Software, La Jolla California, USA). A test was considered statistically significant at a probability of <5% ($p < 0.05$) and we did not assume a Gaussian distribution. Data are represented as mean +/− standard deviation (SD).

# Data availability

The datasets produced in this study are available in the following databases: BioStudies: Images and flow cytometry files; S-BIAD1497.

The source data of this paper are collected in the following database record: biostudies:S-SCDT-10_1038-S44319-025-00383-y.

# Peer review information

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

## Acknowledgements

We thank the technical support from the Biomedical Research Centre core facility members Michael Williams (AbLab) and Andy Johnson (ubcFLOW cytometry), Takahide Murakami (genotyping), Krista Ranta, Joanna Wong, Marcelo Paixao, and Wei Yuan (Vivarium). We acknowledge that the land on which this work was conducted is the traditional, ancestral, and unceded territory of the xwməθkwəy̓əm (Musqueam) People. This work is supported by Canadian Institutes of Health Research (CIHR, PJT-156235 and CIHR PJT-148681) and in part by the Stemcell Network International Travel Award (2022), which enabled its presentation at the FASEB Skeletal Muscle Stem Cells and Regeneration Conference. The Graphical Abstract was created with BioRender.com/t99m665.

## Author contributions

**Melina Messing**: Conceptualization; Data curation; Formal analysis; Supervision; Validation; Investigation; Visualization; Writing—original draft; Project administration; Writing—review and editing. **Marine Theret**: Conceptualization; Resources; Data curation; Formal analysis; Supervision; Validation; Investigation; Methodology; Writing—review and editing. **Michael R Hughes**: Conceptualization; Supervision; Project administration; Writing—review and editing. **Jiaqi Wu**: Data curation; Formal analysis; Validation; Visualization. **Omar Husain Syed**: Data curation; Formal analysis; Validation; Visualization. **Fang Fang Li**: Data curation; Formal analysis. **Yicong Li**: Data curation; Project administration; Writing—review and editing. **Fabio M V Rossi**: Conceptualization; Supervision; Funding acquisition; Writing—review and editing. **Kelly M McNagny**: Conceptualization; Supervision; Funding acquisition; Investigation; Project administration; Writing—review and editing.

Source data underlying figure panels in this paper may have individual authorship assigned. Where available, figure panel/source data authorship is listed in the following database record: biostudies:S-SCDT-10_1038-S44319-025-00383-y.

## Disclosure and competing interests statement

The authors declare no competing interests.

# Expanded View Figures

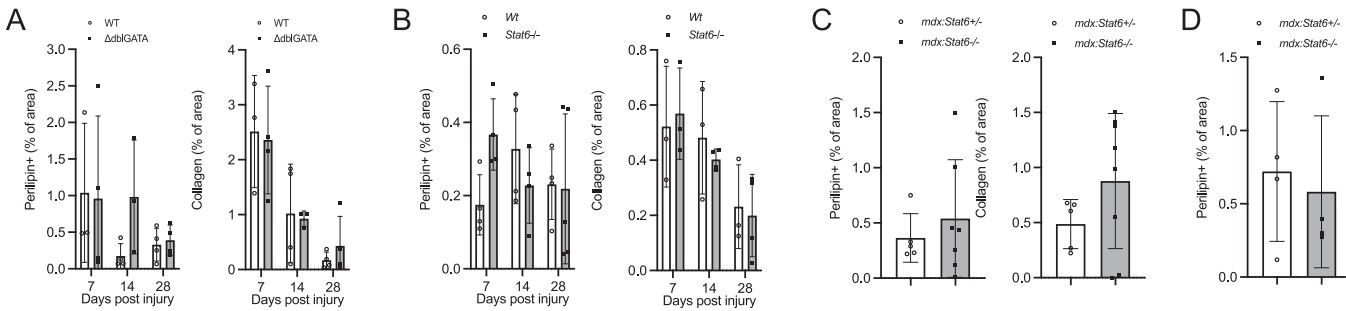

**Figure EV1. Adipocyte and collagen deposition analysis.**

Quantification of % perilipin and % collagen in TA muscle sections *Tibialis anterior* (TA) muscles of wild-type (*Wt*) or ΔdblGATA mice at 7-, 14-, and 28-days post injury after muscle injury with barium chloride (BaCl$_2$) (**A**). Quantification of % perilipin and % collagen in TA muscle sections muscles of wild-type (*Wt*) or *Stat6−/−* mice at 7-, 14-, and 28-days post injury after muscle injury with barium chloride (BaCl$_2$) (**B**). At age 3 months, the right TA of *mdx:Stat6+/−* and *mdx:Stat6−/−* mice were micro-damaged (MD).Tissues were collected at 14 weeks of age. Quantification of % perilipin and % collagen in MD TA muscle sections (**C**). Quantification of % perilipin in 10 months old *mdx:Stat6+/−* and *mdx:Stat6−/−* mice (**D**). Number of mice per group: *n* = 3–4. Error bars represent SD.

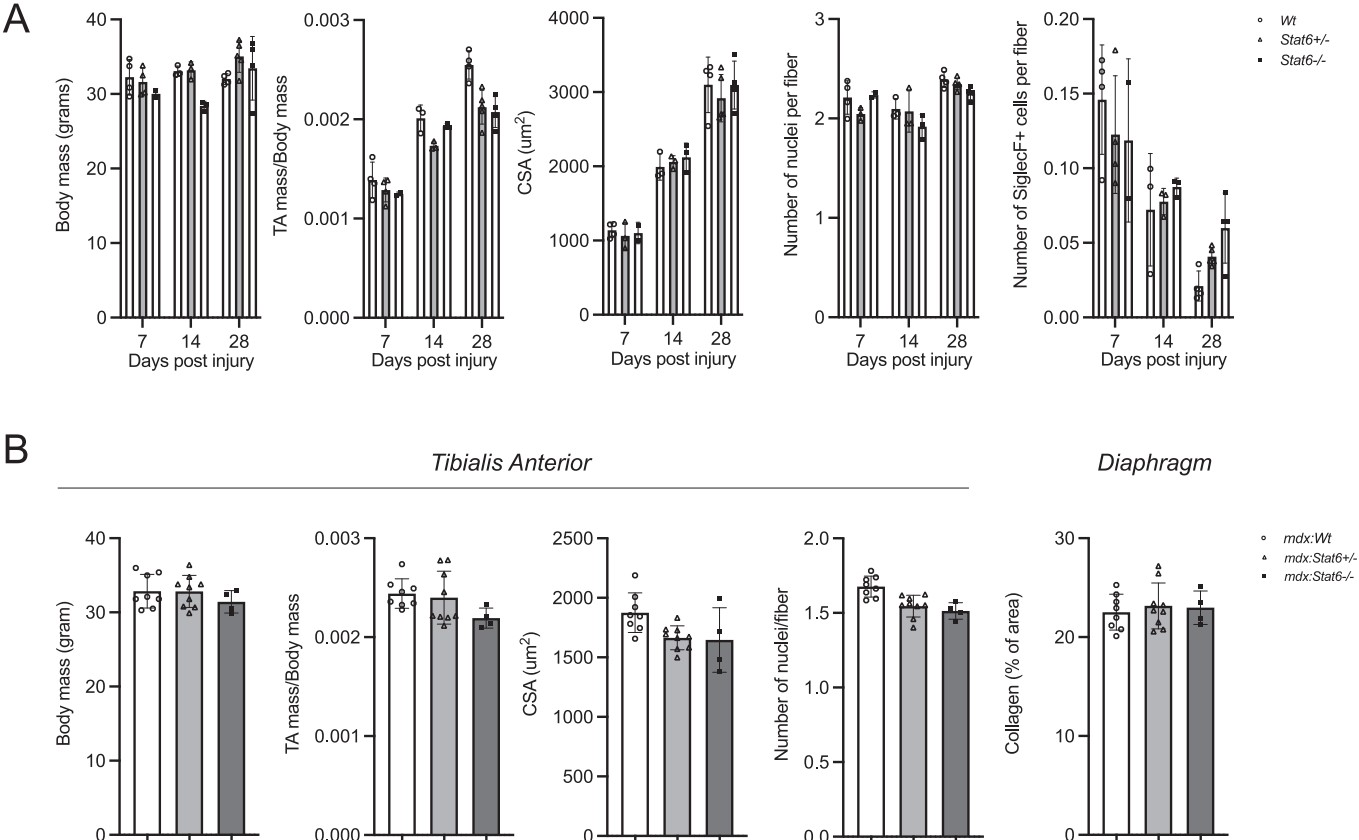

**Figure EV2. Analysis of intermediate loss of STAT6.**

*Tibialis anterior* (TA) muscles of wild-type (*Wt*), *Stat6+/−* or *Stat6−/−* mice were injured with barium chloride (BaCl₂) and body mass, TA mass (normalized to body mass), cross-sectional area (CSA), number of nuclei per fiber and number of SiglecF+ cells was quantified to assess muscle regeneration at 7-, 14-, and 28-days post injury (A). At age 3 months, mdx, *mdx:Stat6+/−* and *mdx:Stat6−/−* mice were compared in terms of body mass, TA/body mass, CSA and nuclei per fiber as well as % collagen deposition in the diaphragm (B). Number of mice per group: *n* = 3–7. Error bars represent SD.

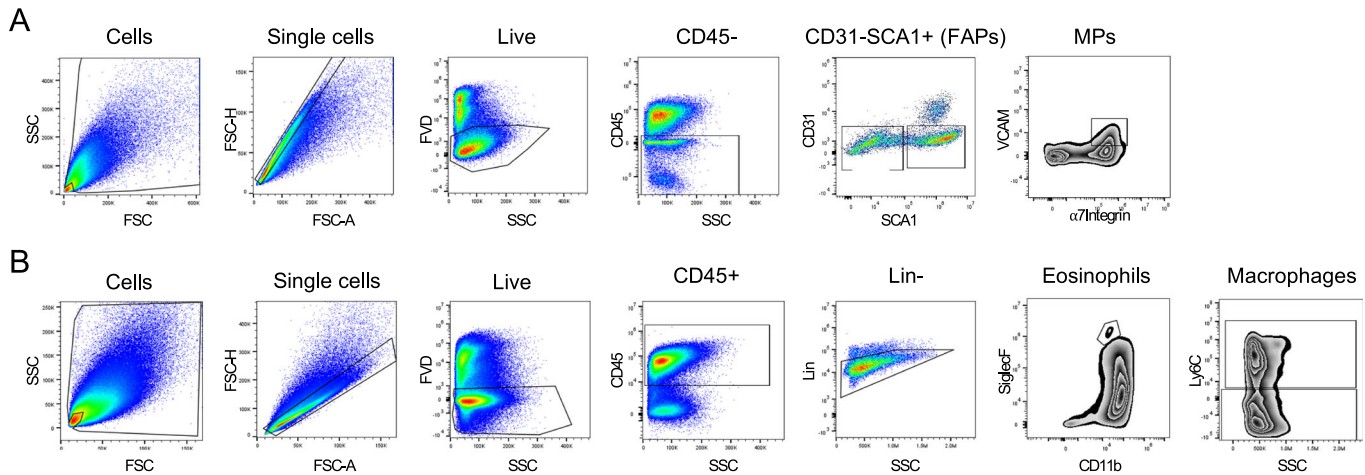

**Figure EV3. Representative flow cytometry gating strategies.**

Muscle resident cells gating strategy (**A**). Eosinophil and macrophage gating strategy (**B**). FVD fixable viability dye, Lin lineage.

