## [Peer Review File · EMBO Reports]

Type-2 innate signals are dispensable for skeletal muscle regeneration and pathology linked to DMD

Melina Messing, Marine Theret, Michael Hughes, Jiaqi Wu, Omar Husain Syed, Fang Fang Li, Yicong Li, Fabio Rossi, and Kelly M. McNagny

Corresponding author(s): Kelly M. McNagny (Kelly@brc.ubc.ca) , Fabio Rossi (fabio@brc.ubc.ca)

Review Timeline:

Submission Date:	28th Apr 24
Editorial Decision:	26th Jun 24
Revision Received:	17th Nov 24
Editorial Decision:	8th Jan 25
Revision Received:	12th Jan 25
Accepted:	21st Jan 25

Editor: Esther Schnapp

Transaction Report:

Dear Kelly,

Thank you for the submission of your manuscript to EMBO reports. We have now received the full set of referee reports that is pasted below.

As you will see, all referees acknowledge that your findings are interesting and should be published. However, the referees also have several suggestions for how the study could be improved, and I would like to know whether you can address all of them? We can discuss the exact revision requirements further, also in a video chat, if you cannot or prefer not to address all comments. For example, I don't know how difficult it is to analyse the effect of eosinophil depletion or inhibiting STAT6 signaling in mouse models of muscle adipogenesis, as referee 2 suggests.

I am confident that we will be able to agree on a specific set of revisions, and I would thus like to invite you to revise your manuscript with the understanding that the referee concerns must be fully addressed and their suggestions taken on board. Please address all referee concerns in a complete point-by-point response. Acceptance of the manuscript will depend on a positive outcome of a second round of review. It is EMBO reports policy to allow a single round of major revision only and acceptance or rejection of the manuscript will therefore depend on the completeness of your responses included in the next, final version of the manuscript.

We realize that it is difficult to revise to a specific deadline. In the interest of protecting the conceptual advance provided by the work, we recommend a revision within 3 months (26th Sep 2024). Please discuss the revision progress ahead of this time with the editor if you require more time to complete the revisions.

- 1) A data availability section providing access to data deposited in public databases is missing. If you have not deposited any data, please add a sentence to the data availability section that explains that.
- 2) Your manuscript contains statistics and error bars based on $n=2$. Please use scatter blots in these cases. No statistics should be calculated if $n=2$.

3) We replaced Supplementary Information with Expanded View (EV) Figures and Tables that are collapsible/expandable online. A maximum of 5 EV Figures can be typeset. EV Figures should be cited as 'Figure EV1, Figure EV2' etc... in the text and their respective legends should be included in the main text after the legends of regular figures.

5) a complete author checklist, which you can download from our author guidelines

<<https://www.embopress.org/page/journal/14693178/authorguide>>. Please insert information in the checklist that is also reflected in the manuscript. The completed author checklist will also be part of the RPF.

6) Please note that all corresponding authors are required to supply an ORCID ID for their name upon submission of a revised manuscript (<<https://orcid.org/>>). Please find instructions on how to link your ORCID ID to your account in our manuscript tracking system in our Author guidelines

<<https://www.embopress.org/page/journal/14693178/authorguide#authorshipguidelines>>

10) Regarding data quantification (see Figure Legends:

<https://www.embopress.org/page/journal/14693178/authorguide#figureformat>)

- the name of the statistical test used to generate error bars and P values,
- the number (n) of independent experiments (please specify technical or biological replicates) underlying each data point,
- the nature of the bars and error bars (s.d., s.e.m.),
- If the data are obtained from $n < 3$, please use scatter blots showing all individual data points.

12) All Materials and Methods need to be described in the main text using our 'Structured Methods' format, which is required for all research articles. According to this format, the Methods section includes a Reagents and Tools Table (listing key reagents, experimental models, software and relevant equipment and including their sources and relevant identifiers) followed by a Methods and Protocols section describing the methods using a step-by-step protocol format. The aim is to facilitate adoption of the methodologies across labs. More information on how to adhere to this format as well as a downloadable template (.docx) for the Reagents and Tools Table can be found in our author guidelines:

An example of a Method paper with Structured Methods can be found here: <https://www.embopress.org/doi/full/10.1038/s44320-024-00037-6#sec-4>

I look forward to seeing a revised form of your manuscript when it is ready.

Referee #1:

In the work by Messing et al, the authors re-investigate the role of eosinophils in skeletal muscle repair using two different murine model, one deficient in eosinophils (Δ dblGATA) and one deficient in IL-4/IL-13 signaling (Stat6^{-/-}). The authors demonstrated with strong evidence that eosinophils and IL-4/IL-13 signaling are not necessary for muscle regenerative processes and are not involved in Duchenne muscular dystrophy pathophysiology, counteracting previous available data. The work is well structured and written.

However, I have some annotations:

in the line 48-50, the authors cited the type 1/2/3-immune responses. I suggest to better specify the most important features of these 3 major types of innate and adaptive cell-mediated effector immunity, as highlighted for example in (10.1016/j.jaci.2014.11.001; J Allergy Clin Immunol. 2015 Mar;135(3):626-35. doi: 10.1016/j.jaci.2014.11.001.)

in the main figures 1 and 2, in the panel A and D the pictures related to T0 for both animal models is missing, as well as the experimental evidences concerning the body mass, TA mass/body mass, the cross sectional area (CSA), the number of SiglecF⁺ cells per fibers. These data are fundamental to better evaluate the modifications of muscular phenotype/architecture in pathological mice after the muscular injury.

Similarly, in the last chapters of the results, the authors compare mdx:Stat6^{-/-} mice to mdx:Stat6^{+/-}. I suggest showing also the differences between mdx mice and mdx:Stat6^{-/-} mice, in order to better understand the effects of the complete ablation of IL-4/IL-13 signaling in mice.

In the end, I have a curiosity. In a recently published paper from Jiang (Ital J Pediatr. 2023; 49: 83. 10.1186/s13052-023-01483-y), in a retrospective cohort study performed on 145 DMD patients between January 2012 and December 2020 the authors demonstrated that eosinophil count in peripheral blood was correlated with the severity of DMD. They suggest that these phenomena could indicate the therapeutic efficacy and prognosis of DMD patients to a certain extent, since eosinophils may be a potentially valuable biomarker or therapeutic target for DMD. Have you ever try to analyze the eosinophil count in peripheral blood of model mice and following the muscular injury?

Referee #2:

In their manuscript entitled, "Type-2 innate signaling is dispensable for skeletal muscle regeneration and pathology linked to Duchenne muscular dystrophy", Messing et al. set out to define the role of eosinophils and IL-4/IL-13 signaling in muscle regeneration and Duchenne muscular dystrophy (DMD). Using genetic approach to deplete eosinophils (dblGATA) or inhibit STAT6 signaling, they convincingly draw the conclusion that these two aspects of type 2 immunity do not influence muscle regeneration following acute injury nor the severity of muscle fibrosis in the mdx mouse model of DMD.

Overall, the experiments are well-designed to broadly assess the impact on muscle regeneration and fibrosis. Although the data are largely negative, it is still an important body of work to publish giving the complex role of type 2 innate immunity in tissue repair and fibrotic disease. The major concern with the manuscript is the over-generalized conclusion that type-2 innate immunity has no role when only two aspects (i.e. eosinophil and STAT6 signaling) of this complex immune response have been examined in this study. M2 macrophages, mast cells, ILC2s and Th2s are cells associated with type 2 immunity and have not

been addressed in this study. IL-4 can also activate PI3K and p38; these pathways were also not examined in the current study.

A further limitation of the study is an examination that was limited to regeneration and fibrosis. Other pathophysiological attributes of muscular dystrophy have not been assessed; membrane stability/repair, calcium homeostasis, and adipogenesis. The latter being particularly important given that prior studies have shown that eosinophils and type 2 cytokines inhibit adipogenic differentiation.

Before the manuscript can be considered further for publication, the authors need to revise the title of the manuscript to reflect what was studied (eosinophils and STAT6-signaling). Second, definitive statements of type 2 immunity and IL-4/IL-13 signaling throughout the manuscript need to be revised accordingly. Third, an examination of the effect of eosinophil depletion or inhibiting STAT6 signaling in mouse models of muscle adipogenesis (B6A/J mouse or glycerol-induced injury) is needed.

Further comments below:

1. The following statement does not take into consideration that other signal transduction pathways are activated downstream of the IL-4 and IL-13 receptor. In summary, we conclude that IL-4/IL-13 signaling is dispensable and does not play a significant role in muscle regeneration, inflammatory cell infiltration or muscle resident cell proliferation in response to injury. Limit the statement to STAT6 signaling.
2. Indicate in the figure legend the age of the mice when they underwent microdamage. How old were they when microdamage was initiated.
3. Some figure panels are not cited in the results (e.g. Figure 3E). Interestingly, a CSA analysis showed a small but significant reduction in the number of small fiber (CSA = 0-400 μm) in MD TAs of mdx:Stat6^{-/-} compared to mdx:Stat6^{+/-}.
4. A comparison of total STAT6 levels, phosphorylation levels and/or expression of STAT6 targets has not been done between STAT6 ^{+/+} and STAT6^{+/-} to rule out a gene dose effect (i.e. STAT6 are the same between STAT6 ^{+/+} and STAT6^{+/-} or is there an 'intermediate' loss of STAT6 that confounds the data).
5. 'Strings' appears to be a typo in line 283-284, linked to this hyper-responsive background may not be broadly extrapolatable to other mouse strings or human disease
6. This following is another example of overly interpreted statement, "type-2 innate signaling is likely not a relevant pathway for the development of targeted immunotherapeutics.
M2 macrophages and ILC2s are cellular components of the type 2 innate immunity, and they produce factors such as IGF1 and amphiregulin, which were not evaluated in this study. Thus, it is not appropriate to broadly conclude that type-2 innate signaling is likely not relevant in DMD. This can be easily addressed by limiting statements to what was specifically manipulated in this study (i.e. eosinophil and STAT6 signaling are likely not relevant).

Referee #3:

The manuscript titled Type-2 innate signaling is dispensable for skeletal muscle regeneration and pathology linked to Duchenne muscular dystrophy described the absence of defect caused by the suppression of eosinophil and IL4/IL13 signalling during muscle regeneration.

The authors show here, the dispensability of eosinophil/ IL4/IL13 signalling during regeneration in opposition with what Heredia et al. suggested in 2013. This previous work proposed that eosinophils plays a role in muscle repair by inhibiting FAPs differentiation through IL4/IL13 signalling (Heredia et al., 2013). However this new work from Messing et al. brings opposite evidence showing that muscle repair in healthy and mdx context exerts no differences when eosinophils number is decrease or when IL4/IL13 signalling is downregulated.

For that purpose, the authors used different mice model, interestingly those models were also used in Heredia et al.,2013. The absence of eosinophils was visited with $\Delta\text{dbpGATA}$ eosinophil-deficient mice as well as Stat6^{-/-} mice where IL4/IL13 signalling was inhibited.

Surprisingly in both of those model WT or mdx background, the respective, deficiency in eosinophils and inhibition of Stat6 did not impact muscle regeneration. First, the authors presented regeneration characteristics of WT $\Delta\text{dbpGATA}$ eosinophil-deficient mouse, where they validate a major decrease in eosinophil number without any significative impact on the regeneration. The same analysis structure has been reproduce using Stat6^{-/-} mice where the same absence of effect has been shown. Messing et al., then proposed to investigated the importance of eosinophils or their signalling during pathological long-term inflammation/regeneration by crossing Stat6^{-/-} mice with mdx background as well as using an $\alpha\text{-IL5}$ neutralizing antibody to prevent differentiation and survival of eosinophils. Even if the mdx phenotype did not appeared very strong, the authors did not identify any major changes during muscle regeneration. The additional injury by micro damaging the tissue as well as the aging process, are respectively in line with the previous observations and showed no differences with their respective controls. In this work Messing et al. brings strong evidences that eosinophils deficiency as well as the inhibition of their major downstream signalling does not affects muscle regeneration.

Overall, the work is well written and use a variety of different techniques that strengthens author's conclusion. I would also like to emphasize the structuration of the paper which helps with the understanding. The work provides good evidences and discussion in order to re-think the importance of eosinophil and their signalling during muscle regeneration. Hereafter are few comments that can clarify and maybe strengthens some of the conclusions.

Comments:

1. Overall the complete absence of effect on muscle regeneration is very surprising based on the impressive phenotype that Heredia et al. showed in their paper. One of the hypothesis proposed by the authors is the difference in term of mouse background between Balb/cJ and C57BL/6.
 - a. To what extent the authors believe that the mouse background has an importance on the phenotype differences? Do you have any supplementary data on Balb/c background which would support this hypothesis?
2. Based on the hypothesis that eosinophils might affect preferentially FAPs fate during muscle regeneration, are you able to quantify the fibrotic deposition in your different regenerative assay like the authors did in Figure 4G-H? Some of your images might show a slight increase which may be lowered because of C57BL/6 background.
3. According to literature, eosinophils is one of the first population recruited on site (\approx 36h). It might be a good idea, to match your flow cytometry data, to add histological images at 3dpi or at an early time point to show a potential defect in early regeneration which might be compensate later at 7dpi by the global immune response.
4. In Figure 1F-G and 2F-G, this is surprising that only 50% of MP or FAPs are incorporating EdU especially at 3dpi. How can you explain the low amount of proliferating MP? Could that come from your previous gating VCAM/ITGa7 where VCAM signal seems dim? To strengthens the conclusion about MP and FAPs proliferation it might be interesting to show another way of quantification. Histological sections and a quantification of Pax7+ and PDGFRa+ cells numbers respectively can be implemented. EdU detection can also be done directly on the section after in vivo injection.
5. For both WT and mdx can the authors show the downregulation of Stat6 protein? Is Stat6^{-/-} model a full body knock out? Can the defect in Stat6 expression affect MP and/or FAPs proliferation/fate independently of IL4/IL13 signalling? Does these populations have the same pattern of proliferation/differentiation in vitro than their respective control? Does this mice have a particular phenotype?
6. Globally, it would be nice if the author can present a WT control along with mdx mice to highlight the mdx phenotype. In this work, mdx background is used as chronic inflammatory model but no increase in eosinophils or macrophages number has been shown compare to the WT condition in Figure 2E and 3E.
7. According to the previous comment, the addition of an extra damage in mdx makes a lot of sense. Why the authors used a mild microdamage injury which might not trigger fully the inflammatory response and eosinophils activation (Desguerre et al., 2012)? Did the authors have data with stronger injury (cardiotoxin or BaCl₂)? Do you think that the slight effect seen on fibers size might be more significant following that type of injury?
8. In Figure 3 and 4, in order to assess the regeneration process in MD mdx, α -IL5 treated mice and aging assay, it would be nice to show a quantification of MP and FAPs numbers, like the authors did in Figure 1 and 2.

Minor comments

1. The flow cytometry data presentation can be improve, please do not use density plot and add the used fluoropher next to the target.
2. In Figure 2d, the representative image of Stat6^{-/-} at 28d staining might be improve in order to show the significant increase.

Reviewer comments and responses:

We thank the reviewers for their thoughtful review and comments on our manuscript. Below, we provide point-by-point responses to the reviews and have revised the manuscript accordingly. All major revisions in the manuscript are highlighted in red.

Referee #1:

In the work by Messing et al, the authors re-investigate the role of eosinophils in skeletal muscle repair using two different murine model, one deficient in eosinophils (Δ dblGATA) and one deficient in IL-4/IL-13 signaling (Stat6^{-/-}). The authors demonstrated with strong evidence that eosinophils and IL-4/IL-13 signaling are not necessary for muscle regenerative processes and are not involved in Duchenne muscular dystrophy pathophysiology, counteracting previous available data. The work is well structured and written.

However, I have some annotations:

A- in the line 48-50, the authors cited the type 1/2/3-immune responses. I suggest to better specify the most important features of these 3 major types of innate and adaptive cell-mediated effector immunity, as highlighted for example in (10.1016/j.jaci.2014.11.001; J Allergy Clin Immunol. 2015 Mar;135(3):626-35. doi: 10.1016/j.jaci.2014.11.001.)

Thank you for this thoughtful suggestion. We have now adjusted the text, accordingly, starting on line 45.

B- in the main figures 1 and 2, in the panel A and D the pictures related to T0 for both animal models is missing, as well as the experimental evidences concerning the body mass, TA mass/body mass, the cross sectional area (CSA), the number of SiglecF+ cells per fibers. These data are fundamental to better evaluate the modifications of muscular phenotype/architecture in pathological mice after the muscular injury.

We have discussed this carefully among the investigators and in consultation with the EMBO Reports Handling Editor. Knowing that there was a **lack of a clear phenotype post injury in our animal models**, we agreed collectively that it is highly unlikely that T0 will be different between animal models. Finally, because we are using a well-established injury model for which **all** timepoints have been published previously we noted that the T0 time point will likely not provide further information beyond what has been previously published (see Jung et. al. reference below). We sincerely hope the reviewer will agree that in the face of previous publications and the challenges linked with the timely evaluation of this control, our existing data is sufficient to support the conclusions of the manuscript.

Jung HW, Choi JH, Jo T, Shin H, Suh JM. Systemic and Local Phenotypes of Barium Chloride Induced Skeletal Muscle Injury in Mice. Ann Geriatr Med Res. 2019 Jun;23(2):83-89. doi: 10.4235/agmr.19.0012. Epub 2019 Jun 30. PMID: 32743293; PMCID: PMC7387593.

C- Similarly, in the last chapters of the results, the authors compare mdx:Stat6^{-/-} mice to mdx:Stat6^{+/-}. I suggest showing also the differences between mdx mice and mdx:Stat6^{-/-} mice, in order to better understand the effects of the complete ablation of IL-4/IL-13 signaling in mice.

We have now added Expanded View Figure 2 which shows a comparison of Wt vs Stat6^{+/-} vs Stat6^{-/-} for the acute model as well as mdx vs mdx:Stat6^{+/-} vs mdx:Stat6^{-/-}. We find no difference with regards to partial or complete ablation of STAT6 and muscle regeneration. We have added this additional information to the text (lines 172 and 233).

D- In the end, I have a curiosity. In a recently published paper from Jiang (Ital J Pediatr. 2023; 49: 83. 10.1186/s13052-023-01483-y), in a retrospective cohort study performed on 145 DMD patients between January 2012 and December 2020 the authors demonstrated that eosinophil count in peripheral blood was correlated with the severity of DMD. They suggest that these phenomena could indicate the therapeutic efficacy and prognosis of DMD patients to a certain extent, since eosinophils may be a potentially valuable biomarker or therapeutic target for DMD. Have you ever try to analyze the eosinophil count in peripheral blood of model mice and following the muscular injury?

Thank you for this intriguing question and the supporting reference (we actually have now added this to the manuscript as reference 41). We indeed examined this for mice at 4, 6 and 8 months of age (with a limited number of n/group). In our hands, blood eosinophils increase from age 4 months to age 6 months but then again drop at 8 months. Thus, in mice, this does not seem to not correlate well with disease progression. See the graph below:

Referee #2:

In their manuscript entitled, "Type-2 innate signaling is dispensable for skeletal muscle regeneration and pathology linked to Duchenne muscular dystrophy", Messing et al. set out to define the role of eosinophils and IL-4/IL-13 signaling in muscle regeneration and Duchenne muscular dystrophy (DMD). Using genetic approach to deplete eosinophils (dblGATA) or inhibit STAT6 signaling, they convincingly draw the conclusion that these two aspects of type 2 immunity do not influence muscle regeneration following acute injury nor the severity of muscle fibrosis in the mdx mouse model of DMD.

A-Overall, the experiments are well-designed to broadly assess the impact on muscle regeneration and fibrosis. Although the data are largely negative, it is still an important body of work to publish giving the complex role of type 2 innate immunity in tissue repair and fibrotic disease. The major concern with the manuscript is the over-generalized conclusion that type-2 innate immunity has no role when only two aspects (i.e. eosinophil and STAT6 signaling) of this complex immune response have been examined in this study. M2 macrophages, mast cells, ILC2s and Th2s are cells associated with type 2 immunity and have not been addressed in this study. IL-4 can also activate PI3K and p38; these pathways were also not examined in the current study.

Understood. We actually structured our manuscript in this way to match the previous Cell paper's phrasing and framing of their work. Indeed, we did not address many aspects of the type-2 innate immune cascade (nor did they). We have now adjusted the text in the manuscript to be more specific with respect to the pathways examined and our conclusions. In discussing this with the EMBO Handling Editor we agreed that, as a good compromise, it would be best to keep the title as is to match the Cell paper and provide a more granular explanation of the reviewers important point in the abstract and text of the manuscript. We have also now added a number of references (see below and references 16-19 in the manuscript) that show that the activation of Th2 immunity is dependent on STAT6. With respect to PI3K and p38 signalling we note that the published Cell manuscript also failed to see these alternative pathways activated by STAT6. Hence, for all the reasons above, we decided to keep our focus on STAT6-mediated TH2 immune response.

Kaplan MH, Schindler U, Smiley ST, Grusby MJ. *Stat6 is required for mediating responses to IL-4 and for development of Th2 cells. Immunity.* 1996;4:313–319. doi: 10.1016/s1074-7613(00)80439-2

Akimoto T, Numata F, Tamura M, Takata Y, Higashida N, Takashi T, Takeda K, et al. *Abrogation of bronchial eosinophilic inflammation and airway hyperreactivity in signal transducers and activators of transcription (STAT)6-deficient mice. J. Exp. Med.* 1998;187:1537–1542. doi: 10.1084/jem.187.9.1537

Akimoto T, Numata F, Tamura M, Takata Y, Higashida N, Takashi T, Takeda K, et al. *Abrogation of bronchial eosinophilic inflammation and airway hyperreactivity in signal transducers and activators of transcription (STAT)6-deficient mice. J. Exp. Med.* 1998;187:1537–1542. doi: 10.1084/jem.187.9.1537

Takeda K, Tanaka T, Shi W, Matsumoto M, Minami M, Kashiwamura S, Nakanishi K, et al. *Essential role of Stat6 in IL-4 signalling. Nature.* 1996;380:627–630. doi: 10.1038/380627a0.

B- A further limitation of the study is an examination that was limited to regeneration and fibrosis. Other pathophysiological attributes of muscular dystrophy have not been assessed; membrane stability/repair, calcium homeostasis, and adipogenesis. The latter being particularly important given that prior studies have shown that eosinophils and type 2 cytokines inhibit adipogenic differentiation.

We were aware of the effects on adipogenic differentiation, but in all of our preliminary experiments we encountered only negative results investigating CSA, N/F, inflammation, and proliferation. We eventually ceased pursuing any of these other avenues and possibilities as we felt that those likely also will not show differences. We have now performed perilipin staining and additional collagen staining on a subset of our animals to probe whether we can see obvious differences in adipocytes and have added these data to a new Expanded View Figure 1. We found that, in line with all other assays performed, there is no striking difference in terms of adipocyte differentiation during muscle regeneration in our models.

C- Before the manuscript can be considered further for publication, the authors need to revise the title of the manuscript to reflect what was studied (eosinophils and STAT6-signaling). Second, definitive statements of type 2 immunity and IL-4/IL-13 signaling throughout the manuscript need to be revised accordingly. Third, an examination of the effect of eosinophil depletion or inhibiting STAT6 signaling in mouse models of muscle adipogenesis (B6A/J mouse or glycerol-induced injury) is needed.

As stated above, we agree with these points and, in consultation with the Editor, have revised the essential elements of text accordingly and have performed perilipin staining to assess adipogenesis. The addition of a new mouse model was deemed to be outside of the scope of this work and not warranted given that we do not see any differences in perilipin+ cell deposition and other aspects of muscle regeneration.

D- Further comments below:

1. The following statement does not take into consideration that other signal transduction pathways are activated downstream of the IL-4 and IL-13 receptor. In summary, we conclude that IL-4/IL-13 signaling is dispensable and does not play a significant role in muscle regeneration, inflammatory cell infiltration or muscle resident cell proliferation in response to injury. Limit the statement to STAT6 signaling.

We have now modified the text accordingly with regards to the significance of these other pathways. As stated above, we noted that the previous Cell manuscript also found no evidence of either PI3K or p38 pathway activation by IL4R/IL13R signalling.

2. Indicate in the figure legend the age of the mice when they underwent microdamage. How old were they when microdamage was initiated.

The mice were 12 weeks of age when the microdamage was started and 14 weeks of age when tissues were collected. We have now added this to the figure legend.

3. Some figure panels are not cited in the results (e.g. Figure 3E). Interestingly, a CSA analysis showed a small but significant reduction in the number of small fiber (CSA = 0-400 μ m) in MD TAs of mdx:Stat6^{-/-} compared to mdx:Stat6^{+/-}.

Thank you for pointing this out. We also added this to the text (line 170).

4. A comparison of total STAT6 levels, phosphorylation levels and/or expression of STAT6 targets has not been done between STAT6 ^{+/+} and STAT6^{+/-} to rule out a gene dose effect (i.e. STAT6 are the same between STAT6 ^{+/+} and STAT6^{+/-} or is there an 'intermediate' loss of STAT6 that confounds the data).

We have added Expanded View Figure 2 which shows a comparison of Wt vs Stat6^{+/-} vs Stat6^{-/-} for the acute model as well as mdx vs mdx:Stat6^{+/-} vs mdx:Stat6^{-/-}. We show that there is no difference with regards to partial or complete ablation of STAT6 and muscle regeneration. We have now added this in the text accordingly (lines 172 and 233). Due to an absence of and "intermediate" loss effect of STAT6 deletion, levels of phosphorylation have not been investigated.

5. 'Strings' appears to be a typo in line 283-284, linked to this hyper-responsive background may not be broadly extrapolatable to other mouse strings or human disease.

Apologies. This has been corrected.

6. This following is another example of overly interpreted statement, "type-2 innate signaling is likely not a relevant pathway for the development of targeted immunotherapeutics. M2 macrophages and ILC2s are cellular components of the type 2 innate immunity, and they produce factors such as IGF1 and amphiregulin, which were not evaluated in this study. Thus, it is not appropriate to broadly conclude that type-2 innate signaling is likely not relevant in DMD. This can be easily addressed by limiting statements to what was specifically manipulated in this study (i.e. eosinophil and STAT6 signaling are likely not relevant).

We have adjusted the text accordingly (line 348) to be more moderate and narrower with regards to the conclusions on this point.

Referee #3:

The manuscript titled Type-2 innate signaling is dispensable for skeletal muscle regeneration and pathology linked to Duchenne muscular dystrophy described the absence of defect caused by the suppression of eosinophil and IL4/IL13 signaling during muscle regeneration.

The authors show here, the dispensability of eosinophil/ IL4/IL13 signaling during regeneration in opposition with what Heredia et al. suggested in 2013. This previous work proposed that eosinophils plays a role in muscle repair by inhibiting FAPs differentiation through IL4/IL13 signaling (Heredia et al., 2013). However this new work from Messing et al. brings opposite evidence showing that muscle repair in healthy and mdx context exerts no differences when eosinophils number is decreased or when IL4/IL13 signaling is downregulated. For that purpose, the authors used different mice models, interestingly those models were also used in Heredia et al., 2013. The absence of eosinophils was visited with Δ dblGATA eosinophil-deficient mice as well as Stat6^{-/-} mice where IL4/IL13 signaling was inhibited. Surprisingly in both of those model WT or mdx background, the respective deficiency in eosinophils and inhibition of Stat6 did not impact muscle regeneration. First, the authors presented regeneration characteristics of WT Δ dblGATA eosinophil-deficient mouse, where they validate a major decrease in eosinophil number without any significative impact on the regeneration. The same analysis structure has been reproduced using Stat6^{-/-} mice where the same absence of effect has been shown. Messing et al., then proposed to investigated the importance of eosinophils or their signaling during pathological long-term inflammation/regeneration by crossing Stat6^{-/-} mice with mdx background as well as using an α -IL5 neutralizing antibody to prevent differentiation and survival of eosinophils. Even if the mdx phenotype did not appear very strong, the authors did not identify any major changes during muscle regeneration. The additional injury by micro damaging the tissue as well as the aging process, are respectively in line with the previous observations and showed no differences with their respective controls. In this work Messing et al. brings strong evidence that eosinophils deficiency as well as the inhibition of their major downstream signaling does not affect muscle regeneration.

Overall, the work is well written and use a variety of different techniques that strengthens author's conclusion. I would also like to emphasize the structuration of the paper which helps with the understanding. The work provides good evidences and discussion in order to re-think the importance of eosinophil and their signalling during muscle regeneration. Hereafter are few comments that can clarify and maybe strenghtens some of the conclusions.

Comments:

1. Overall the complete absence of effect on muscle regeneration is very surprising based on the impressive phenotype that Heredia et al. showed in their paper. One of the hypothesis proposed by the authors is the difference in term of mouse background between Balb/cJ and C57BL/6.

a. To what extent the authors believe that the mouse background has an importance on the phenotype differences? Do you have any supplementary data on Balb/c background which would support this hypothesis?

We have now performed acute muscle injury in BALB/c mice compared to C57BL/6 mice and have now added this data as an additional main figure, Figure 5. We have evaluated body mass, TA mass (normalized to BM), CSA, N/F, adipocyte accumulation, collagen deposition and number of SiglecF⁺ cells. We have also added these results to the text starting in line 238. The results indicate that there is indeed a difference in the ability of these two strains to regenerate muscle after injury. The CSA analysis indicates a delay in regeneration in the BALB/c mice and we have also observed a trend toward more adipocyte accumulation (not significant) and significantly more collagen deposition at day 28 after injury. Further, we observed significantly more SiglecF⁺ cells (eosinophils) at 7 days post injury. We think this demonstrate that even without any other genetic manipulation, the two strains already show differences which may be exacerbated by the absence of eosinophils or IL-4 signaling, perhaps to the point of an inability of BALB/c

muscle to regenerate entirely (as suggested by Heredia et al.). We do not have the required mouse strains to explore this further (i.e., all the mutant strains on a BALB/c background), but we feel this additional data strengthens our hypothesis. This underscores that utilizing various mouse models with different genetic backgrounds, as done by Heredia et al., likely complicates the interpretation of results and introduces confusion regarding the roles of immune cells and pathways studied.

2. Based on the hypothesis that eosinophils might affect preferentially FAPs fate during muscle regeneration, are you able to quantify the fibrotic deposition in your different regenerative assay like the authors did in Figure 4G-H? Some of your images might show a slight increase which may be lowered because of C57BL/6 background.

We have now added collagen quantifications of a subset of our animals to Expanded View Figure 1. We observe no significant difference in collagen deposition across all models tested (beyond the differences we see on the BALB/c background mentioned above).

3. According to literature, eosinophils is one of the first population recruited on site (≈ 36 h). It might be a good idea, to match your flow cytometry data, to add histological images at 3dpi or at an early time point to show a potential defect in early regeneration which might be compensated later at 7dpi by the global immune response.

We were unable to generate mice to perform histology at 3dpi in a timely manner. However, we argue that even if there is a defect in early regeneration, it is not significant enough to cause long term effects (as suggested by Heredia et. al, who argue that eosinophils/IL-4 signaling is essential for muscle regeneration). The lack of differences between both number and proliferation of MPs and FAPs at 3dpi supports the conclusion that that there are no impactful differences in regeneration even at an earlier time point.

4. In Figure 1F-G and 2F-G, this is surprising that only 50% of MP or FAPs are incorporating EdU especially at 3dpi. How can you explain the low amount of proliferating MP? Could that come from your previous gating VCAM/ITGa7 where VCAM signal seems dim? To strengthen the conclusion about MP and FAPs proliferation it might be interesting to show another way of quantification. Histological sections and a quantification of Pax7+ and PDGFRa+ cells numbers respectively can be implemented. EdU detection can also be done directly on the section after *in vivo* injection.

As mentioned above, we were not able to generate additional mice to perform further measurements (such as Pax7+ or PDGFRa+ staining) at 3dpi. However, when comparing our results to previously published work (Hardy et al., 2016), we note that the % of proliferating MPs seems to be typical for the injury model that we used (BaCl₂). In the reference below (Hardy et al, 2016), about 50% of MPs are Ki67+ at 4dpi (as shown in Figure 1). Since, beyond a lack of differences in proliferation, we also do not observe a difference in regeneration and the % proliferation which matches the previous work, we are confident in these results and hope that the reviewer will agree.

Hardy D, Besnard A, Latil M, Jouvion G, Briand D, Thépenier C, Pascal Q, Guguin A, Gayraud-Morel B, Cavaillon JM, Tajbakhsh S, Rocheteau P, Chrétien F. Comparative Study of Injury Models for Studying Muscle Regeneration in Mice. *PLoS One*. 2016 Jan 25;11(1):e0147198. doi: 10.1371/journal.pone.0147198. PMID: 26807982; PMCID: PMC4726569.

5. For both WT and mdx can the authors show the downregulation of Stat6 protein? Is Stat6^{-/-} model a full body knock out? Can the defect in Stat6 expression affect MP and/or FAPs proliferation/fate independently of IL4/IL13 signalling? Does these populations have the same pattern of proliferation/differentiation *in vitro* than their respective control? Does this mice have a particular phenotype?

The model we are using is a full body KO and these mice have no phenotype at steady state. When challenged with some form of type-2 immune inducing agent (various allergens for example), these mice do show defects in the ability to mount an appropriate type-2 immune response (see references below and reference 16-19 in the manuscript). Since we do not observe any obvious phenotype in the absence of STAT6 in our mice, we do not believe that STAT6 expression affects MPs or FAPs. If there is an effect, it is likely too subtle to detect. We have also added an additional Expanded View Figure 2 that does explore Wt, vs HEMI vs full KO of STAT6 and show that a partial loss of STAT6 also does not affect muscle regeneration nor confound our findings.

Kaplan MH, Schindler U, Smiley ST, Grusby MJ. *Stat6* is required for mediating responses to IL-4 and for development of Th2 cells. *Immunity*. 1996;4:313–319. doi: 10.1016/s1074-7613(00)80439-2

Akimoto T, Numata F, Tamura M, Takata Y, Higashida N, Takashi T, Takeda K, et al. Abrogation of bronchial eosinophilic inflammation and airway hyperreactivity in signal transducers and activators of transcription (STAT)6-deficient mice. *J. Exp. Med*. 1998;187:1537–1542. doi: 10.1084/jem.187.9.1537

Akimoto T, Numata F, Tamura M, Takata Y, Higashida N, Takashi T, Takeda K, et al. Abrogation of bronchial eosinophilic inflammation and airway hyperreactivity in signal transducers and activators of transcription (STAT)6-deficient mice. *J. Exp. Med.* 1998;187:1537–1542. doi: 10.1084/jem.187.9.1537

Takeda K, Tanaka T, Shi W, Matsumoto M, Minami M, Kashiwamura S, Nakanishi K, et al. Essential role of Stat6 in IL-4 signalling. *Nature.* 1996;380:627–630. doi: 10.1038/380627a0.

6. Globally, it would be nice if the author can present a WT control along with mdx mice to highlight the mdx phenotype. In this work, mdx background is used as chronic inflammatory model but no increase in eosinophils or macrophages number has been shown compare to the WT condition in Figure 2E and 3E.

This comparison has been recently published by the laboratory of Dr. Armando Villalta (see reference below Kastenschmidt et al). In this work, the authors compared eosinophil counts in Wt compared to mdx mice and show that there is a consistent elevation of eosinophils at 4, 12 and 52 weeks of age. We have added a sentence our manuscript pointing this out and also added the very relevant reference (line 284, reference 38).

Kastenschmidt JM, Coulis G, Farahat PK, Pham P, Rios R, Cristal TT, Mannaa AH, Ayer RE, Yahia R, Deshpande AA, Hughes BS. A stromal progenitor and ILC2 niche promotes muscle eosinophilia and fibrosis-associated gene expression. *Cell reports.* 2021 Apr 13;35(2).

7. According to the previous comment, the addition of an extra damage in mdx makes a lot of sense. Why the authors used a mild microdamage injury which might not trigger fully the inflammatory response and eosinophils activation (Desguerre et al., 2012)? Did the authors have data with stronger injury (cardiotoxin or BaCl₂)? Do you think that the slight effect seen on fibers size might be more significant following that type of injury?

Our goal was to induce persistent chronic damage and fibrosis deposition (rather than a one-time acute injury reminiscent of the injection of BaCl₂). While we do not have the data requested, we show that with advanced disease (aged mdx mice), even though there is significantly more inflammation and fibrosis, there continues to be no phenotypical changes in STAT6^{+/+} vs STAT6^{-/-} mdx mice.

8. In Figure 3 and 4, in order to assess the regeneration process in MD mdx, α -IL5 treated mice and aging assay, it would be nice to show a quantification of MP and FAPs numbers, like the authors did in Figure 1 and 2.

Unfortunately, at the time of the experiment for the α -IL5 treated mice, we did not record the number of FAPs and MPs. With regard to the aged mdx mice experiment, since there are no changes in collagen deposition, or myofiber CSA, we hypothesised that there would not be any changes in MP and FAP number. In addition, even if there were changes, we have shown that it has no effect on muscle histopathology.

Minor comments

1. The flow cytometry data presentation can be improve, please do not use density plot and add the used fluoropher next to the target.

We have added the fluorophore to the images. We, respectfully, would like to keep the density plot style as it as it better represents cell population separation.

2. In Figure 2d, the representative image of Stat6^{-/-} at 28d staining might be improve in order to show the significant increase.

We have adjusted the image accordingly.

Dear Kelly, and happy new year !

Thank you for the submission of your revised manuscript. We have now received the enclosed reports from the referees and I am happy to say that all support its publication now. Only a few editorial requests will need to be addressed before we can proceed with the official acceptance of your manuscript:

- Your ms has 5 main figures but is laid out as a full article with separate results and discussion sections. Please either add one more main figure or combine the results and discussion sections to publish your paper as a short report. The character count will need to be reduced a little, which usually occurs naturally when results and discussion are combined.
- Please add up to 5 keywords to your ms file.
- Please add the direct URL for the deposited dataset to the DAS.
- Please rename the conflict of interest subheading to "Disclosure Statement and Competing Interests"
- Please remove the author credits from the ms file. All credits are entered during online ms submission.
- Please correct the REFERENCE FORMAT to the EMBO reports format: It needs to be alphabetical, not numerical; et al needs to be used after 10 author names; DOIs should only be used for preprints and datasets that have not been published yet
- The author checklist Data Availability section has not been completed. Please send us a new completed checklist.
- The FUNDING INFO needs to be part of the Acknowledgments.
- Please add a callout for Fig. 5E.
- Materials and Methods should be "Methods".
- Our systematic figure check of accepted ms detected a possible partial image reuse within Fig 1A (2 images on the left) and a potential full image reuse between Fig 1D (WT) and Fig 2D (STAT6^{-/-}). Can you please explain what happened?

Figure Legends - Comments

- Please note that the legend for figure EV 1d is missing in the manuscript. This needs to be rectified.
- Please note that the legend for figures 1f-g; 2f-g are mislabeled as 1d-e; 2f-g in the manuscript. This needs to be rectified.
- Please note that the exact p values are not provided in the legends of figures 1d-e; 2d; 3b, d-h; 4a-c, e-f, h-i; 5c-e.
- Please note that in figures 1d-e; 2d; 4a-c, e-f, h-i; 4a-c, e-f, h-i; 5c-e; there is a mismatch between the annotated p values in the figure legend and the annotated p values in the figure file that should be corrected.
- Please note that the error bars are not defined in the legends of figures 1b-g, 2b-g; 3a-b, d-h; 4a-c, e-f, h-i; 5b-e; EV 1a-d; EV 2a-b.

I would like to suggest some changes to the abstract that needs to be written in present tense:

Immune responses play an integral role in skeletal muscle regeneration. In the genetically inherited muscle disease Duchenne muscular dystrophy (DMD), muscle regeneration is disrupted, leading to chronic inflammation, fibrosis, and early mortality. Previously, it has been suggested that type-2 innate immune cells, particularly eosinophils and their production of IL-4, play an essential role in effective muscle regeneration after acute injury. We here re-investigate the role of eosinophils in skeletal muscle repair using mice deficient in eosinophils (Δ dblGATA), or deficient in IL-4R/IL-13R signaling through STAT6 (Stat6^{-/-}). We show that neither deficiency has an impact on skeletal muscle regeneration in response to acute injury as quantified by fiber size, immune cell infiltration, or muscle-resident stem cell proliferation. We also investigate the role of STAT6 signaling in mdx:Stat6^{-/-} mice, a model of DMD and, again, find that ablation of STAT6 signaling has no effect on the rate or severity of fibrotic scar formation or disease progression. In contrast to previous models, our data suggest a negligible role for eosinophils and STAT6 signaling in skeletal muscle regeneration after acute or chronic injury.

I think the title is fine, given that "Type-2 innate signaling" as meant here is specified in the abstract.

EMBO press papers are accompanied online by A) a short (1-2 sentences) summary of the findings and their significance, B) 2-3 bullet points highlighting key results and C) a synopsis image that is exactly 550 pixels wide and 200-600 pixels high (the

height is variable). The synopsis image should provide a sketch of the major findings, like a graphical abstract. Please note that text needs to be readable at the final size. Please send us this information along with the final manuscript.

Best,
Esther

Referee #1:

According to the concerns raised in the last revision, the authors improved significantly the manuscript.

Referee #2:

While the authors' response adequately addresses most of my concerns, I still do not agree with the title in the current form. We run the risk that readers will make hasty conclusions by reading the title of the manuscript, without a thorough evaluation of the great work presented in this manuscript or abstract. Given that this manuscript has not evaluated all elements of type 2 innate immunity, we run a dangerous course in which the field may abandon future studies aimed at exploring facets of type 2 immunity that have been explored in the current manuscript. I strongly recommend a revision of the title, whose final acceptance I defer to the discretion of the editors.

Referee #3:

The authors have satisfactorily addressed the issues raised by the reviewers.

1. Your ms has 5 main figures but is laid out as a full article with separate results and discussion sections. Please either add one more main figure or combine the results and discussion sections to publish your paper as a short report. The character count will need to be reduced a little, which usually occurs naturally when results and discussion are combined. **We decided to add one additional figure (Figure 3 was split into Figures 3 and 4).**
2. Change title re latest Editor Email. **Completed**
3. Please add up to 5 keywords to your ms file. **Completed**
4. Please add the direct URL for the deposited dataset to the DAS. **Completed**
5. Please rename the conflict of interest subheading to "Disclosure Statement and Competing Interests" **Completed**
6. Please remove the author credits from the ms file. All credits are entered during online ms submission. **Completed**
7. Please correct the REFERENCE FORMAT to the EMBO reports format: It needs to be alphabetical, not numerical; et al needs to be used after 10 author names; DOIs should only be used for preprints and datasets that have not been published yet **Completed**
8. The author checklist Data Availability section has not been completed. Please send us a new completed checklist. **Completed**
9. The FUNDING INFO needs to be part of the Acknowledgments. **Completed**
10. Please add a callout for Fig. 5E. **Completed**
11. Materials and Methods should be "Methods". **Completed**
12. Our systematic figure check of accepted ms detected a possible partial image reuse within Fig 1A (2 images on the left) and a potential full image reuse between Fig 1D (WT) and Fig 2D (STAT6^{-/-}). Can you please explain what happened? **We carefully reviewed the figures but do not notice an accidental duplication between the indicated groups/images. Is it possible that this can happen when the phenotypes are very similar?**

Figure Legends - Comments

13. Please note that the legend for figure EV 1d is missing in the manuscript. This needs to be rectified. **Completed**
14. Please note that the legend for figures 1f-g; 2f-g are mislabeled as 1d-e; 2f-g in the manuscript. This needs to be rectified. **Completed**

15. Please note that the exact p values are not provided in the legends of figures 1d-e; 2d; 3b, d-h; 4a-c, e-f, h-i; 5c-e. **Completed**
16. Please note that in figures 1d-e; 2d; 4a-c, e-f, h-i; 4a-c, e-f, h-i; 5c-e; there is a mismatch between the annotated p values in the figure legend and the annotated p values in the figure file that should be corrected. **Completed**
17. Please note that the error bars are not defined in the legends of figures 1b-g, 2b-g; 3a-b, d-h; 4a-c, e-f, h-i; 5b-e; EV 1a-d; EV 2a-b. **Completed**
18. I would like to suggest some changes to the abstract that needs to be written in present tense:

Immune responses play an integral role in skeletal muscle regeneration. In the genetically inherited muscle disease Duchenne muscular dystrophy (DMD), muscle regeneration is disrupted, leading to chronic inflammation, fibrosis, and early mortality. Previously, it has been suggested that type-2 innate immune cells, particularly eosinophils and their production of IL-4, play an essential role in effective muscle regeneration after acute injury. We here re-investigate the role of eosinophils in skeletal muscle repair using mice deficient in eosinophils (Δ dblGATA), or deficient in IL-4R/IL-13R signaling through STAT6 (Stat6^{-/-}). We show that neither deficiency has an impact on skeletal muscle regeneration in response to acute injury as quantified by fiber size, immune cell infiltration, or muscle-resident stem cell proliferation. We also investigate the role of STAT6 signaling in mdx:Stat6^{-/-} mice, a model of DMD and, again, find that ablation of STAT6 signaling has no effect on the rate or severity of fibrotic scar formation or disease progression. In contrast to previous models, our data suggest a negligible role for eosinophils and STAT6 signaling in skeletal muscle regeneration after acute or chronic injury. **Completed**

19. EMBO press papers are accompanied online by A) a short (1-2 sentences) summary of the findings and their significance, B) 2-3 bullet points highlighting key results and C) a synopsis image that is exactly 550 pixels wide and 200-600 pixels high (the height is variable). The synopsis image should provide a sketch of the major findings, like a graphical abstract. Please note that text needs to be readable at the final size. Please send us this information along with the final manuscript. **Completed**

Dr. Kelly M. McNagny
British Columbia, University of
The Biomedical Research Centre
University of British Columbia
2222 Health Sciences Mall
Vancouver, British Columbia V6T 1Z3
Canada

Dear Kelly,

I am very pleased to accept your manuscript for publication in the next available issue of EMBO reports. Thank you for your contribution to our journal.

Best,
Esther
